



# 1 Calibrating Networks of Low-
# 2 Cost Air Quality Sensors

Priyanka deSouza[1*], Ralph Kahn[2], Tehya Stockman[3,4], William Obermann[3], Ben Crawford[5], An
Wang[6], James Crooks[7,8], Jing Li[9], Patrick Kinney[10]
1: Department of Urban and Regional Planning, University of Colorado Denver, 80202
2: NASA Goddard Space Flight Center, Greenbelt MD
3: Denver Department of Public Health and Environment, USA
4: Department of Civil, Environmental, and Architectural Engineering, University of Colorado
Boulder, Boulder, Colorado 80309, United States
5: Department of Geography and Environmental Sciences, University of Colorado Denver, 80202
6: Senseable City Lab, Massachusetts Institute of Technology, Cambridge 02139
7: Division of Biostatistics and Bioinformatics, National Jewish Health, 2930
8: Department of Epidemiology, University of Colorado at Denver - Anschutz Medical Campus,
15 129263
9: Department of Geography and the Environment, University of Denver, Denver, CO, USA
10: Boston University School of Public Health, Boston, MA, USA
*: priyanka.desouza@ucdenver.edu

## 20 Abstract

Ambient fine particulate matter (PM$_{2.5}$) pollution is a major health risk. Networks of low-cost
sensors (LCS) are increasingly being used to understand local air pollution variation. However,
measurements from LCS have uncertainties which can act as a potential barrier for effective
decision-making. LCS data thus need to be calibrated to obtain better quality PM$_{2.5}$ estimates. In
order to develop correction factors, LCS are typically co-located with gold-standard reference
monitors. A calibration equation is then developed that relates the raw output of the LCS as closely
as possible to measurements from the reference monitor. This calibration algorithm is then
typically *transferred* to measurements from monitors in the network. Calibration algorithms tend to
be evaluated based on their performance at co-location sites. It is often implicitly assumed that the
conditions at the relatively sparse co-location sites are representative of the LCS network, overall.
Little work has been done to explicitly evaluate the sensitivity of the LCS network hotspot
detection, and spatial and temporal PM$_{2.5}$ trends to the correction method applied. This paper
provides a first look at how transferable different calibration methods are using a dense network of
Love My Air LCS monitors in Denver. It offers a series of transferability metrics that can be
applied to other networks and offers suggestions for which calibration method would be most
useful for different end goals. Finally, it develops a set of best practice suggestions on calibrating
LCS networks.





**Key words**: low-cost sensors, PM$_{2.5}$, calibration, Love My Air

# 1 Introduction

Poor air quality is currently the single largest environmental risk factor to human health in the
world, with ambient air pollution responsible for 6.7 million premature deaths every year (State of
Global Air, 2020). Accurate air quality data is crucial for tracking long-term trends in air quality
levels, and for the development of effective pollution management plans. Levels of fine particulate
matter (PM$_{2.5}$), a criterion pollutant that poses more of danger to human health than other
widespread pollutants (Kim et al., 2015), can vary over distances as small as ~ 10's of meters in
complex urban environments (Brantley et al., 2019; deSouza et al., 2020a). Therefore, dense
monitoring networks are often needed to capture relevant spatial variations. Due to their costliness,
EPA air quality reference monitoring networks, the gold standard for measuring air pollutants, are
sparsely positioned across the US (Apte et al., 2017; Anderson and Peng, 2012).
Low-cost sensors (LCS) (<USD $2500 as defined by the US EPA Air Sensor Toolbox) (Williams
et al., 2014) have the potential to capture concentrations of PM in previously unmonitored
locations and democratize air pollution information (Castell et al., 2017; Kumar et al., 2015;
Morawska et al., 2018; Snyder et al., 2013; deSouza and Kinney, 2021; deSouza, 2022). However,
LCS measurements have several sources of uncertainty (Bi et al., 2020; Giordano et al., 2021;
Liang, 2021).
Most low-cost PM sensors rely on optical measurement techniques. Optical instruments face
several inherent challenges that introduce potential differences in mass estimations compared to
reference methods (Barkjohn et al., 2021; Crilley et al., 2018; Giordano et al., 2021; Malings et al.,
62 2020):

1. Optical methods do not directly measure mass concentrations; rather, they estimate mass based
on calibrations that convert light scattering data to particle number and mass. LCS come with
factory-supplied calibrations, but in practice must be re-calibrated in the field to ensure accuracy,
due to variations in ambient particle characteristics.
2. High relative humidity (RH) can produce hygroscopic particle growth, leading to mass
overestimation if the particles are not dessicated by the instrument.
3. The inability to detect particles with diameters below a specific size, which is determined by
the wavelength of laser light within each device, and is generally in the vicinity of 0.3 μm, whereas
the peak in pollution particle size distributions is typically smaller than 0.3 μm.
4. The physical and chemical parameters of the aerosol (particle size distribution, shape, indices
of refraction, hygroscopicity, volatility etc.) which might vary significantly across different
microenvironments with diverse sources impact light scattering, which in turn affects the aerosol
mass concentrations reported by these instruments.





The need for field calibration to correct LCS measurements is particularly important. This is
typically done by co-locating a small number of LCS with a reference monitor at a representative
monitoring location or locations. The co-location could be carried out for a brief period before
and/or after the actual study or may continue at a small number of sites for the duration of the
study. In either case, the co-location provides data from which a calibration equation is then
developed that relates the raw output of the LCS as closely as possible to the desired quantity as
measured by the reference monitor. Thereafter, the calibration equation is transferred to other LCS
in the network, based upon the presumption that ongoing sampling conditions are within the same
range as those during the calibration period.

Calibration models typically correct for 1) systematic error in LCS by adjusting for bias using
reference monitor measurements, and 2) the dependence of LCS measurements on environmental
conditions affecting the ambient particle properties such as relative humidity (RH), temperature
(T), and/or dew-point (D). Correcting for RH, T and D is carried out through either a) a physics-
based approach that accounts for aerosol hygroscopic growth given particle composition using $\kappa$-
kohler's theory, or b) empirical models, such as regression and machine learning techniques. In
this paper, we will focus on the latter, as it is the most widely used (Barkjohn et al., 2021).
Previous work has also shown that the two approaches yield comparable improvements in the case
of PM$_{2.5}$ LCS (Malings et al., 2020).

Prior studies have used multivariate regressions, piecewise linear regressions, or higher-order
polynomial models to account for RH, T and D in these calibration equations (Holstius et al., 2014;
Magi et al., 2020; Zusman et al., 2020). More recently, machine learning techniques such as
random forests, neural networks, and gradient boosted decision trees have been used (Considine et
al., 2021; Liang, 2021; Zimmerman et al., 2018). Researchers have also started including
additional covariates in their models besides what is directly measured by the LCS, such as time of
day, seasonality and site-type, which have been shown to yield significantly improved results
(Considine et al., 2021).

Past research has shown that there are several important decisions, in addition to the choice of
statistical model, that need to be made during calibration and can impact the results (Bean, 2021;
Giordano et al., 2021; Hagler et al., 2018). These include a) the kind of reference air quality
monitor used, b) the time-interval (e.g., hour/day) over which to average measurements used when
developing the calibration algorithm, c) how cross-validation (e.g., leave one site out/10-fold cross
validation) is carried out, and d) how long the co-location experiment takes place.

Calibration algorithms are evaluated based on how well the corrected measurements agree with
those from the reference monitor. A commonly used metric is the coefficient of determination, $R^2$,
which quantifies the strength of the association. However, it might be a mis-leading indicator of
sensor performance when measurements are observed close to the level of detection of the
instrument. Therefore, Root Mean Square Error (RMSE) is also often used in practice. Neither of
these metrics captures how well the calibration method developed at the co-located sites *transfers*
to the rest of the network.



If the conditions at the calibration site (meteorological conditions, pollution source mix) are the
same as at the rest of the network, the calibration function developed at the co-location site can be
assumed to be transferable to the rest of the network. In order to ensure that the sampling
conditions of the co-location site are representative of sampling conditions of the network, most
researchers tend to deploy monitors in the same general sampling area as the network (Zusman et
al., 2020). However, it is difficult to definitively test if the co-location site is representative of the
locations of all monitors in the network; ambient PM concentrations can vary on scales as small as
a few meters. Furthermore, LCS are often deployed specifically in areas where the air pollution
conditions are poorly understood, meaning that representativeness cannot be assessed ahead-of-
time.
Where multiple co-location sites exist, one way to address this challenge is to leave out one or
another co-location site to test if the calibration algorithm is transferable to the left-out site. This
method was used in recent work evaluating the feasibility of developing a US-wide correction to
the PurpleAir low-cost sensor network (Barkjohn et al., 2021; Nilson et al., 2022). Although this
approach helps, co-location sites are sparse relative to other sites in the network. Even in the
PurpleAir network (which is one of the densest low-cost networks in the world) there were only 39
co-location sites in 16 US states, a small fraction of the several thousand PurpleAir sites overall
(Barkjohn et al., 2021). It is thus important to test how sensitive the spatial and temporal trends of
pollution derived from the network are to the calibration algorithm used.
Examining the reliability of calibration methods is timely because, as mentioned earlier, more
researchers are opting to use machine learning calibration models. Although in most cases, such
models have yielded better results than traditional linear regressions, it is important to examine if
these models are overfitted to conditions at the co-location sites, and how transferable they are to
the rest of the network. Indeed, because of concerns of overfitting, some researchers have
explicitly eschewed employing machine learning calibration models altogether (Nilson et al.,
2022). It is important to test if these concerns are warranted.
This paper uses a dense low-cost air quality monitoring network deployed in Denver, termed
"Love My Air" network, to quantify the uncertainty in the spatial and temporal trends of the
network to the calibration algorithm used, as well as to ask the question: How much do we have to
worry about the transferability of different calibration functions across a $PM_{2.5}$ network in a
relatively small area in a single city? The methodology proposed in this paper to evaluate the
transferability of calibration adjustments can be applied to other low-cost sensor networks, with the
understanding that the actual results will vary with study region.
## 2 Data and Methods
### 2.1 Data Sources
Between January 1 and September 30, 2021, Denver's Love My Air sensor network collected data
from 24 low-cost sensors deployed across the city outside of public schools and at reference



monitor locations (**Figure 1, Table 1**). The Love My Air sensors are Canary-S models equipped
with a Plantower 5003, made by Lunar Outpost Inc. The Canary-S sensors detect $PM_{2.5}$, T, and
RH, and upload minute-resolution measurements to an online platform via cellular data network.

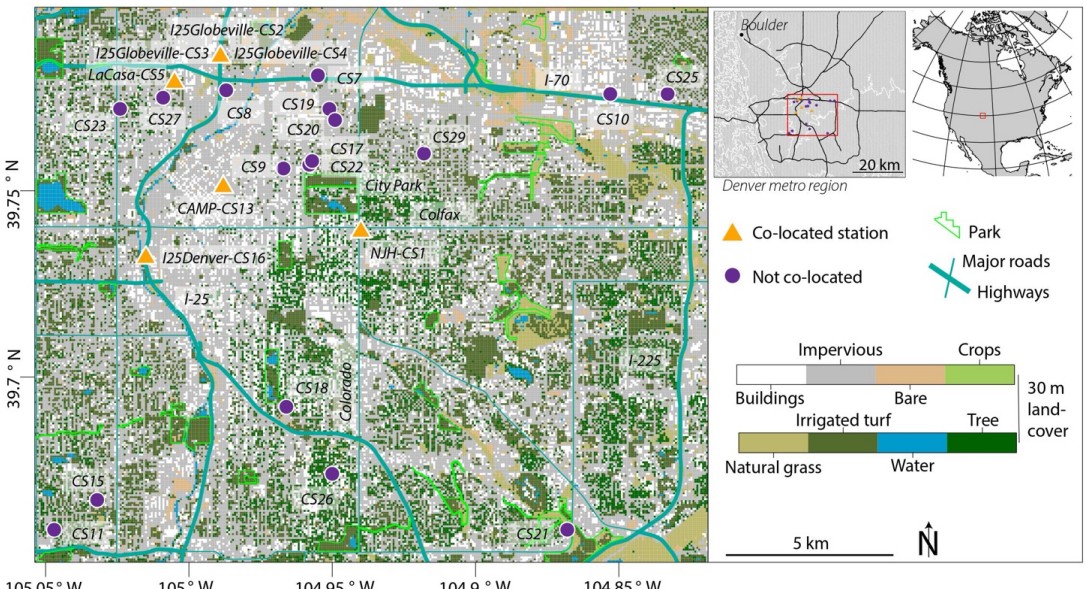


*Figure 1: Locations of all 24 Love My Air Sensors. Sensors displayed with an orange triangle*
*indicate that they were co-located with a reference monitor. The labels of the co-located sensors include the*
*name of the corresponding reference monitor. The base map of land cover was obtained from*
*https://drcog.org/services-and-resources/data-maps-and-modeling/regional-land-use-land-cover-*
*project, last accessed April 2021.*

After removing missing values in the $PM_{2.5}$, T and RH data, RH < 0 (unrealistic values), T ≤ -30$^0$C
(unrealistically low), and $PM_{2.5}$ values above 1,500 μg/m³ (outside the operational range of the
Plantower sensors used) from the Canary-S sensors (Considine et al., 2021), we were left with
8,809,340 measurements. We calculated hourly averages and obtained a total 147,101
measurements. From inspection, one of the monitors, CS13, worked intermittently in January and
February, before resuming continuous measurement in March (**Figure S1** *in Supplementary*
*Information*). When CS13 worked intermittently, large spikes in the measurements were observed,
likely due to power surges. We thus only retained measurements taken after March 1, 2021, for this
monitor. The total number of hourly measurements was thus reduced to 146,583.

Love My Air sensors were co-located with FEM reference monitors at La Casa (Sensor ID: CS5),
CAMP (Sensor ID: CS13), I25 Globeville (Sensor ID: CS2, CS3, CS4), I25 Denver (Sensor ID:
CS16), and NJH (Sensor ID: CS1) for the entire period of the experiment. Three Love My Air
sensors were co-located with the I25 Globeville Monitor, whereas there were single Love My Air
sensors at the other co-location sites. We obtained high-quality hourly $PM_{2.5}$ measurements from
the five reference monitors for the duration of the experiment. We joined hourly averages from
each of the co-located Love My Air monitors with the corresponding FEM monitor. We had a total
of 35,593 co-located measurements for which we had data for both the Love My Air sensor and the


corresponding reference monitor. **Figure S2** displays time-series plots of PM$_{2.5}$ from all co-located
Love My Air sensors. **Figure S3** displays time-series plots of PM$_{2.5}$ from the corresponding
reference monitors.
***Table 1***: *Site location of each Love My Air sensor, as well as summary statistics of hourly*
*measurements from each sensor*

| Sensor ID | Co-location Information | Latitude | Longitude | Hours operational | PM$_{2.5}$ (µg/m³) | | | Temperature (⁰C) | RH (%) | Dewpoint (⁰C) |
| | | | | | Mean | Median | Min-Max | Mean | Mean | Mean |
|---|---|---|---|---|---|---|---|---|---|---|
| CS1 | Co-located at NJH | 39.739 | -104.940 | 5,478 | 13 | 8 | 0 - 121 | 14.9 | 57.4 | 4.4 |
| CS2 | Co-located at I25 Globeville | 39.786 | -104.989 | 5,818 | 14 | 9 | 0 - 142 | 16.4 | 63.6 | 7.6 |
| CS3 | Co-located at I25 Globeville | 39.786 | -104.989 | 2,490 | 18 | 13 | 0 - 159 | 9.3 | 62.5 | 0.1 |
| CS4 | Co-located at I25 Globeville | 39.786 | -104.989 | 5,765 | 12 | 8 | 0 - 137 | 15.8 | 67.6 | 8.0 |
| CS5 | Co-located at La Casa | 39.779 | -105.005 | 5,761 | 12 | 8 | 0 - 129 | 13.4 | 69.6 | 6.0 |
| CS7 | - | 39.781 | -104.955 | 6,540 | 13 | 8 | 0 - 136 | 16.5 | 55.6 | 5.0 |
| CS8 | - | 39.777 | -104.987 | 6,282 | 13 | 8 | 0 - 133 | 17.3 | 38.3 | 0.0 |
| CS9 | - | 39.756 | -104.967 | 6,552 | 12 | 8 | 0 - 115 | 15.3 | 62.8 | 6.1 |
| CS10 | - | 39.776 | -104.853 | 6,552 | 12 | 7 | 0 - 142 | 17.9 | 32.6 | -2.4 |
| CS11 | - | 39.659 | -105.047 | 6,548 | 12 | 7 | 0 - 127 | 15.0 | 58.2 | 4.5 |
| CS13 | Co-located at CAMP | 39.751 | -104.988 | 4,449 | 13 | 8 | 0 - 115 | 21.9 | 54.7 | 10.2 |
| CS15 | - | 39.667 | -105.032 | 6,552 | 10 | 6 | 0 - 106 | 17.0 | 34.6 | -1.5 |
| CS16 | Co-located at I25 Denver | 39.732 | -105.015 | 5,832 | 12 | 9 | 0 - 100 | 17.4 | 33.6 | -2.2 |
| CS17 | - | 39.757 | -104.958 | 6,527 | 12 | 7 | 0 - 149 | 17.1 | 35.1 | -1.3 |
| CS18 | - | 39.692 | -104.966 | 6,552 | 12 | 7 | 0 - 115 | 16.9 | 36.3 | -1.0 |
| CS19 | - | 39.772 | -104.951 | 1,749 | 11 | 5 | 0 - 66 | 3.4 | 40.0 | -11.1 |
| CS20 | - | 39.769 | -104.949 | 6,551 | 10 | 6 | 0 - 105 | 17.9 | 34.2 | -1.2 |
| CS21 | - | 39.659 | -104.868 | 6,551 | 12 | 6 | 0 - 129 | 15.2 | 39.2 | -1.2 |
| CS22 | - | 39.758 | -104.957 | 6,551 | 12 | 7 | 0 - 118 | 17.5 | 35.4 | -0.9 |
| CS23 | - | 39.772 | -105.024 | 6,552 | 14 | 9 | 0 - 139 | 16.5 | 34.6 | -2.0 |
| CS25 | - | 39.776 | -104.833 | 6,551 | 12 | 7 | 0 - 135 | 16.2 | 35.8 | -1.8 |
| CS26 | - | 39.674 | -104.950 | 6,552 | 12 | 7 | 0 - 115 | 15.9 | 36.9 | -1.2 |



| CS27 | - | 39.775 | -105.009 | 6,552 | 12 | 7 | 0 - 115 | 16.4 | 35.6 | -1.4 |
| CS29 | - | 39.760 | -104.918 | 6,552 | 11 | 7 | 0 - 114 | 15.7 | 37.5 | -1.2 |


The three Love My Air sensors co-located at the I25 Globeville sites (CS2, CS3, CS4) agreed well
with each other (correlation = 0.98) (**Figures S4** and **Figure S5**). To ensure that our co-located
dataset was well balanced across sites, we only retained measurements from CS2 at the I25
Globeville site. We were left with a total of 27,338 co-located measurements that we used to
develop a calibration algorithm. **Figure S6** displays the time-series plots of PM$_{2.5}$ from all other
Love My Air sensors in the network.

Reference monitors at La Casa, CAMP, I25 Globeville and I25 Denver, also reported minute-level
PM$_{2.5}$ concentrations between April 23 11:16 and September 30, 22:49. We joined minute-level
Love my Air concentrations with minute-level reference data at these sites. We had a total of
1,062,141 co-located minute-level measurements during this time period. As with the hourly-
averaged data, we only retained data from one of the Love My Air sensors at the I25 Globeville
site and were thus left with 815,608 measurements. **Table S1** has information on the minute-level
co-located measurements. **Figure S7** displays the time-series plot of minute-level data from the
LCS at the four co-location sites. As can be seen, the data at the minute-level displays more
variation and peaks in PM$_{2.5}$ concentrations than the hourly-averaged measurements, likely due to
the impact of passing sources. It is also important to mention that minute-level reference data may
have some additional uncertainties given the time resolution. Unless explicitly referenced, we will
be reporting results from using hourly-averaged measurements.

We found that RH and T reported by the Love My Air sensors were well correlated with that
reported by the reference monitoring stations. We used the Love My Air T and RH measurements
in our calibration models as they most closely represent the conditions experienced by the sensors.

We derived dew-point (D) from T and RH reported by the Love My Air sensors using the
*weathermetrics* package in the programming language R (Anderson and Peng, 2012), as D has
been shown to be a good proxy of particle hygroscopic growth in previous research (Clements et
al., 2017; Malings et al., 2020). Some previous work has also used a nonlinear correction for RH in
the form of $RH^2/(1-RH)$, that we also calculated for this study.

We extracted hour, weekend, and month variables from the Canary-S sensors and converted hour
and month into cyclic values to capture periodicities in the data by taking the cosine and sine of
hour*$2\pi/24$ and month*$2\pi/12$, which we designate as cos_time, sin_time, cos_month and
sin_month, respectively. Sinusoidal corrections for seasonality have been shown to improve
accuracy of PM$_{2.5}$ measurements in machine learning models(Considine et al., 2021).

## 2.2 Statistical Modeling

The goal of the calibration algorithm is to predict, as accurately as possible, the 'true' PM$_{2.5}$
concentrations given the concentrations reported by the Love My Air sensors. At the co-located



sites, the FEM PM$_{2.5}$ measurements, which we take to be the "true" PM$_{2.5}$ concentrations, are the
dependent variable in the models. We tested 21 increasingly complex models that included T, RH,
D as well as metrics that captured the time-varying patterns of PM$_{2.5}$ to correct the Love My Air
PM$_{2.5}$ measurements (**Table 2**).

Sixteen models were multivariate models that were used in a recent paper (Barkjohn et al., 2021) to
calibrate another network of low-cost sensors: the PurpleAir, that rely on the same PM$_{2.5}$ sensor
(Plantower) as the Canary-S monitors in this study. As T, RH and D are not independent (**Figure**
**S8**), the 16 linear regression models include adding the meteorological conditions considered as
interaction terms, instead of additive terms. The remaining 5 relied on machine learning
techniques.

Machine learning models can capture more complex nonlinear effects (for instance, unknown
relationships between additional spatial and temporal variables). We opted to use the following
machine learning techniques that have been widely used in calibrating LCS:

1. *Random forest (RF)*: RF is a decision-tree-based machine learning algorithm that has been
shown to perform well in air quality predictions. Briefly, to generate a random forest model, the
user specifies the maximum number of trees that make up the forest. Each tree is constructed using
a bootstrapped random sample from the training data set. The origin node of the decision tree is
split into sub-nodes by considering a random subset of the possible explanatory variables. Trees
are split based on which of the explanatory variables in each subset is the strongest predictor of the
outcome. This process of node splitting is repeated until a terminal node is reached (Zimmerman et
al., 2018). For our random forest models, the terminal node was specified using a minimum node
size of five data points per node.

2. *Neural Network (NN)*: NN consists of interconnected neurons organized in layers. Each neuron
or unit passes received information through an activation function and produces output values that
are then processed by neurons in the next layer. The NN training process is based on updating the
weights of neurons via supervised learning (Spinelle et al., 2014). A simple single hidden layer
neural network with a linear transfer function was chosen in this study.

3. *Gradient Boosting (GB)*: GB is a decision-tree-based approach that uses 'boosting' methods to
improve model performance. 'Boosting' sequentially combines many 'weak' models (learners)
into a final, improved model. The final model is built in an additive forward stagewise manner
where at each step a new learner is added that minimizes the negative gradient using a least squares
approach. The residuals of the current model are then used as the input for the next tree allowing
the model to 'learn' from the errors of the previous models (Johnson et al., 2018).

4. *SuperLearner (SL)*: SL is an ensemble-based machine learning algorithm, which allows for the
simultaneous evaluation (by cross-validation) of a library of plausible machine learning algorithms
to determine which models are most appropriate for the data, based on minimizing a least squares



loss function, and then averages over these chosen models to produce a composite model (Van der Laan et al., 2007).

All machine learning models were run using the *caret* package in R (Kuhn, 2015).

### 2.2.1 Types of Corrections

For each of the 21 models considered, we developed four main corrections:

(C1) Developed using training data for the entire period of co-location.
(C2) Developed using all data for the same week of the measurement.
(C3) Developed using co-located data collected for a brief period (2 weeks) at the beginning of the study (Jan 1 - Jan 14, 2021).
(C4) Developed using co-located data collected for two 2-week periods in different seasons (Jan 1 - Jan 14, 2021, and May 1 - May 14, 2021).

Although models developed using co-located data over the entire time period (C1) tend to be more accurate over the entire spatiotemporal data set, it is inefficient to re-run large models frequently (incorporating new data). On-the-fly corrections (such as C2) can help characterize short-term variation in air pollution and sensor characteristics. The duration of calibration is a key question that remains unanswered (Liang, 2021). We opted to test corrections C3 and C4 as many low-cost sensor networks rely on developing calibration models based on relatively short co-location periods (deSouza et al., 2020b; West et al., 2020; Singh et al., 2021).

### 2.2.2 Cross-Validation techniques to avoid overfitting in the machine learning models

We used a Leave-One-Site (I25 Globeville, I25 Denver, La Casa, CAMP)-Out (LOSO) approach for cross validation (CV) to prevent overfitting in our machine learning models (Models 17 - 21 in **Table 2**). Briefly, we split the data into four groups, with each group excluding data from a single reference monitoring site. In each cross-validation iteration, we selected each group in turn to fit the model and made predictions at the left-out site. This CV approach was used to tune the hyper parameters in the machine learning models adopted in this study using correction approaches: C1, C2, C3 and C4.

For the correction conducted on the complete archived dataset (C1), we also conducted a leave-out-by-date (LOBD) CV for the machine learning models considered (**Table 3**). For the LOBD model validation method, the project time period was split into 3-week periods. Each period contained between ~ 700 and 900 hourly data points, with typically more sensors running continuously during later chunks as more sensors were deployed and came online over time. Thirteen periods were available in total, and, for each test-train set, 12 periods were used to train the correction model, whereas the remaining interval was selected to test the correction model. By eliminating, using data from the same calendar week, where measurements are likely to be correlated, we eliminate the possibility of obtaining overly optimistic model performance summary statistics.





Models were generated for all combinations of training and test data. To summarize: each of the 21
calibration models considered was tested under four potential correction schemes (C1, C2, C3 and
C4). For C1, the machine-learning algorithms were trained using two CV approaches: LOSO and
LOBD, separately. For C2, C3 and C4 only LOSO was conducted, as model application is already
being performed on a different time period from the training. Note that for simple linear
regressions, overfitting is not an issue, and no CV is required.

Zusman et al., (2020) have reported that for more than 3 co-location sites, a LOSO CV is preferred,
as it replicates our ultimate objective of applying the calibration developed to other sites in the
network. However, in this case, due to the high correlation across co-located sites (**Figure S5,**
**Figure S6**), a LOBD CV is likely to produce more robust results.

Overall, we test 89 models (26 (C1) + 21 x 3 (C2, C3, C4) = 89) listed in **Tables 2** and **3**.
**2.2.3 Evaluating the correction models at the co-location sites**
**Figure S9** displays the PM$_{2.5}$ concentrations from the reference monitors and the corresponding
levels from the co-located Love My Air sensors by RH. Uncorrected Love My Air measurements
tend to be biased upwards by an average of ~12%.

We evaluate the performance of the corrections across the range of PM$_{2.5}$ concentrations for the
entire time period of co-location in our sample using the following metrics: R (Pearson correlation
coefficient), and RMSE (**Tables 2** and **3**). We also evaluated calibrations using corrections C3 and
C4 only for the time-period over which the calibration algorithm was developed, which was Jan 1 -
Jan 14, 2021, for C3 and Jan 1 - Jan 14, 2021, and May 1 - May 14, 2021 for C4 (**Table S2**).

Mean PM$_{2.5}$ concentrations from the reference monitors between Jan 1 - Jan 14, 2021, was 9 μg/m$^3$
(Median: 7 μg/m$^3$, Min:0 μg/m$^3$, Max: 79 μg/m$^3$). Nineteen measurements were > 30 μg/m$^3$.  Mean
PM$_{2.5}$ concentrations from the reference monitors between May 1 - May 14 was 6 μg/m$^3$ (Median:
5 μg/m$^3$, Min: 1 μg/m$^3$, Max: 22 μg/m$^3$). Zero measurements were > 30 μg/m$^3$.

We evaluated model performance for true/reference PM$_{2.5}$ concentrations > 30 μg/m$^3$ and ≤ 30
μg/m$^3$, as these concentrations account for the greatest differences in health and air pollution
avoidance behavior impacts (Nilson et al., 2022). Further, lower concentrations (PM$_{2.5}$ ≤ 30 μg/m$^3$)
represent most measurements observed in our network; better performance at these levels will
ensure better day-to-day functionality of the correction. In order to compare errors observed in the
two different concentration ranges, in addition to reporting R and RMSE of the calibration
approaches, we also report the normalized RMSE (normalized by the mean of the true
concentrations) (**Table S3**).

One of the key advantages of LCS is that they report high frequency measurements of pollution.
As reference monitoring stations provide hourly, or daily average pollution values, most often the
calibration algorithm is developed using hourly averaged data and then applied to the high
frequency LCS measurements. We applied the calibration algorithms described in **Tables 2** and **3**



developed using hourly-averaged co-located measurements on minute-level measurements from
the co-located LCS described in **Table S1**. We evaluated the performance of the corrected high-
frequency measurements against the 'true' measurements from the corresponding reference
monitor using the metrics R and RMSE (**Tables 4** and **5**).
***Table 2****: Performance of the calibration models as captured using root mean square error*
*(RMSE), and Pearson correlation (R). LOSO CV was used to prevent overfitting in the machine*
*learning models. All corrected values were evaluated over the entire time-period (Jan 1 -*
*September 30, 2021)*

| ID | Name | Equation | C1 *Correction developed on data during the entire period of network operation* | | C2 *On-the-fly correction developed using data for the same week of measurement* | | C3 *Correction developed using measurements made in the first two weeks of January* | | C4 *Correction developed using measurements from the first two weeks of January and the first two weeks in May* | |
|---|---|---|---|---|---|---|---|---|---|---|
| | | | R | RMSE (µg/m³) | R | RMSE (µg/m³) | R | RMSE (µg/m³) | R | RMSE (µg/m³) |
| | **Raw Love My Air measurements** | | | | | | | | | |
| 0 | Raw | | 0.927 | 6.469 | - | - | - | - | - | - |
| | **Multivariate Regression (LOSO CV)** | | | | | | | | | |
| 1 | Linear | $PM_{2.5, corrected} = PM_{2.5}$ x $s1 + b$ | 0.927 | 3.421 | 0.944 | 3.008 | 0.927 | 3.486 | 0.927 | 3.424 |
| 2 | +RH | $PM_{2.5, corrected} = PM_{2.5}$ x $s_1 + RH$ x $s_2 + b$ | 0.929 | 3.379 | 0.948 | 2.904 | 0.928 | 3.618 | 0.929 | 3.462 |
| 3 | +T | $PM_{2.5, corrected} = PM_{2.5}$ x $s_1 + T$ x $s_2 + b$ | 0.928 | 3.409 | 0.949 | 2.896 | 0.925 | 3.948 | 0.928 | 3.460 |
| 4 | +D | $PM_{2.5, corrected} = PM_{2.5}$ x $s_1 + D$ x $s_2 + b$ | 0.928 | 3.417 | 0.947 | 2.934 | 0.917 | 3.713 | 0.925 | 3.470 |
| 5 | +RH x T | $PM_{2.5, corrected} = PM_{2.5}$ x $s_1 + RH$ x $s_2 + T$ x $s_3 +$ $RH$ x $T$ x $s_4 + b$ | 0.934 | 3.260 | 0.953 | 2.782 | 0.931 | 3.452 | 0.933 | 3.344 |
| 6 | +RH x D | $PM_{2.5, corrected} = PM_{2.5}$ x $s_1 + RH$ x $s_2 + D$ x $s_3 +$ $RH$ x $D$ x $s_4 + b$ | 0.930 | 3.361 | 0.953 | 2.785 | 0.911 | 3.973 | 0.929 | 3.461 |
| 7 | +D x T | $PM_{2.5, corrected} = PM_{2.5}$ x $s_1 + D$ x $s_2 + T$ x $s_3 + D$ | 0.928 | 3.409 | 0.952 | 2.798 | 0.888 | 5.698 | 0.921 | 3.720 |





| | | | | | | | | | | |
|---|---|---|---|---|---|---|---|---|---|---|
| | | x T x $s_4$ + b | | | | | | | | |
| 8 | +RH x T x D | $PM_{2.5, corrected}$ = $PM_{2.5}$ x $s_1$ + RH x $s_2$ + T x $s_3$ + D x $s_4$ + RH x T x $s_5$ + RH x D x $s_6$ + T x D x $s_7$ + RH x T x D x $s_8$ + b | 0.935 | 3.246 | 0.955 | 2.724 | 0.779 | 7.077 | 0.926 | 3.625 |
| 9 | PM x RH | $PM_{2.5, corrected}$ = $PM_{2.5}$ x $s_1$ + RH x $s_2$ + RH x $PM_{2.5}$ x $s_3$ + b | 0.930 | 3.362 | 0.950 | 2.854 | 0.925 | 3.949 | 0.925 | 3.767 |
| 10 | PM x D | $PM_{2.5, corrected}$ = $PM_{2.5}$ x $s_1$ + D x $s_2$ + D x $PM_{2.5}$ x $s_3$ + b | 0.932 | 3.324 | 0.950 | 2.871 | 0.883 | 4.460 | 0.913 | 3.777 |
| 11 | PM x T | $PM_{2.5, corrected}$ = $PM_{2.5}$ x $s_1$ + T x $s_2$ + T x $PM_{2.5}$ x $s_3$ + b | 0.930 | 3.365 | 0.952 | 2.809 | 0.906 | 6.509 | 0.928 | 3.466 |
| 12 | PM x nonlinear RH | $PM_{2.5, corrected}$ = $PM_{2.5}$ x $s_1$ + $\frac{RH^2}{(1-RH)}$ x $s_2$ + $\frac{RH^2}{(1-RH)}$ x $PM_{2.5}$ x $s_3$ + b | 0.934 | 3.277 | 0.948 | 2.900 | 0.931 | 3.510 | 0.932 | 3.403 |
| 13 | PM x RH x T | $PM_{2.5, corrected}$ = $PM_{2.5}$ x $s_1$ + RH x $s_2$ + T x $s_3$ + $PM_{2.5}$ x RH x $s_4$ + $PM_{2.5}$ x T x $s_5$ + RH x T x $s_6$ + $PM_{2.5}$ x RH x T x $s_7$ + b | 0.938 | 3.165 | 0.956 | 2.672 | 0.891 | 6.220 | 0.928 | 3.497 |
| 14 | PM x RH x D | $PM_{2.5, corrected}$ = $PM_{2.5}$ x $s_1$ + RH x $s_2$ + D x $s_3$ + $PM_{2.5}$ x RH x $s_4$ + $PM_{2.5}$ x D x $s_5$ + RH x D x $s_6$ + $PM_{2.5}$ x RH x D x $s_7$ + b | 0.933 | 3.288 | 0.957 | 2.663 | 0.879 | 7.289 | 0.917 | 4.033 |
| 15 | PM x T x D | $PM_{2.5, corrected}$ = $PM_{2.5}$ x $s_1$ + T x $s_2$ + D x $s_3$ + $PM_{2.5}$ x T x $s_4$ + $PM_{2.5}$ x D x $s_5$ + T x D x $s_6$ + $PM_{2.5}$ x T x D x $s_7$ + b | 0.932 | 3.315 | 0.957 | 2.665 | 0.734 | 6.302 | 0.905 | 4.574 |
| 16 | PM x RH x T x D | $PM_{2.5, corrected}$ = $PM_{2.5}$ x $s_1$ + RH x $s_2$ + T x $s_3$ + D x $s_4$ + $PM_{2.5}$ x RH x $s_5$ + $PM_{2.5}$ x T x $s_6$ + T x RH x $s_7$ + $PM_{2.5}$ x D x $s_8$ + D x RH x $s_9$ + D x T x $s_{10}$ + $PM_{2.5}$ x RH x T x $s_{11}$ + $PM_{2.5}$ x RH x | 0.940 | 3.115 | 0.960 | 2.557 | 0.324 | 32.951 | 0.765 | 6.746 |





| | | | | | | | | | | | |
|---|---|---|---|---|---|---|---|---|---|---|---|
| | | D x $s_{12}$ + PM$_{2.5}$ x D x T x $s_{13}$ + D x RH x T x $s_{14}$ + PM$_{2.5}$ x RH x T x D x $s_{15}$ + b | | | | | | | | | |
| | **Machine Learning (LOSO CV)** | | | | | | | | | | |
| 17 | Random Forest | PM$_{2.5, corrected}$ = f(PM$_{2.5}$, T, RH) | 0.983 | 1.713 | 0.988 | 1.450 | 0.913 | 3.926 | 0.911 | 3.824 |
| 18 | Neural Network (One hidden layer) | PM$_{2.5, corrected}$ = f(PM$_{2.5}$, T, RH) | 0.933 | 3.286 | 0.948 | 2.916 | 0.932 | 3.550 | 0.913 | 4.725 |
| 19 | Gradient Boosting | PM$_{2.5, corrected}$ = f(PM$_{2.5}$, T, RH) | 0.950 | 2.870 | 0.964 | 2.452 | 0.910 | 3.854 | 0.909 | 3.834 |
| 20 | SuperLearner | PM$_{2.5, corrected}$ = f(PM$_{2.5}$, T, RH) | 0.950 | 2.855 | 0.970 | 2.236 | 0.910 | 3.917 | 0.923 | 3.582 |
| 21 | Random Forest | For C1: PM$_{2.5, corrected}$ = f(PM$_{2.5}$, T, RH, D, cos_time, cos_month, sin_month)  For C2, C3, C4 PM$_{2.5, corrected}$ = f(PM$_{2.5}$, T, RH, D, cos_time) | 0.987 | 1.475 | 0.990 | 1.289 | 0.870 | 5.032 | 0.884 | 4.617 |


***Table 3**: Performance of the calibration models using the C1 correction as captured using root mean square error (RMSE), normalized RMSE, and Pearson correlation (R) LOBD CV was used to prevent overfitting in the machine learning models*

| ID | Machine Learning (LOBD CV) | | R | RMSE (μg/m³) |
|---|---|---|---|---|
| 17 | Random Forest | PM$_{2.5, corrected}$ = f(PM$_{2.5}$, T, RH) | 0.983 | 1.710 |
| 18 | Neural Network (One hidden layer) | PM$_{2.5, corrected}$ = f(PM$_{2.5}$, T, RH) | 0.933 | 3.285 |
| 19 | Gradient Boosting | PM$_{2.5, corrected}$ = f(PM$_{2.5}$, T, RH) | 0.953 | 2.759 |
| 20 | SuperLearner | PM$_{2.5, corrected}$ = f(PM$_{2.5}$, T, RH) | 0.956 | 2.692 |
| 21 | Random Forest | PM$_{2.5, corrected}$ = f(PM$_{2.5}$, T, RH, D, cos_time, cos_month, sin_month) | 0.987 | 1.480 |





### 2.3 Evaluating transferability

### 2.3.1 Evaluating the representativeness of meteorological conditions at the co-location sites of the entire network

We first evaluated if meteorological conditions (T and RH) at the co-location sites corresponding to measurements used to construct calibration models were representative of conditions of operation for the rest of the network by comparing distributions of these parameters across sites.

### 2.3.2 Evaluating transferability at the co-location sites

To evaluate how transferable the calibration technique developed at the co-located sites was to the rest of the network, we ran the models proposed in **Tables 2** and **3**, after leaving out each one of the 5 co-located sites in turn. We report the distribution of RMSE from each model across the different test datasets using boxplots (**Figure 2**).

We compare statistically the errors in predictions on each test dataset with errors in predictions from using all sites in our main analysis. Such an approach is useful to understand how well the proposed correction can transfer to other areas in the Denver region. To compare statistical difference between errors, t-tests were used to compare normally distributed datasets (as determined by Shapiro–Wilk), and Wilcoxon tests were used for nonparametric datasets with a significance value of 0.05.

We have only 5 co-location sites in the network. Although evaluating the transferability among these sites is useful, as we know the true $PM_{2.5}$ concentrations at these sites, we also evaluated the transferability of these models in the larger network by predicting $PM_{2.5}$ concentrations using the models proposed in **Tables 2** and **3** at each of the 24 sites in the Love My Air network. For each site, we display time series plots of corrected $PM_{2.5}$ measurements in order to visually compare the ensemble of corrected values at each site.

### 2.3.3 Evaluating the sensitivity of hotspot detection across the network of sensors to the calibration method

One of the key use-cases of low-cost sensors is hotspot detection. We report the labels of sites that are the most polluted using corrected measurements from the 89 different models using hourly data. We repeat this process for daily, weekly and monthly-averaged corrected measurements. We ignore missing measurements from the network when calculating time averaged values for the different time periods considered. We report the mean number of sensors that are ranked 'most polluted' across the different correction functions for the different averaging periods.

### 2.3.4 Evaluating sensitivity of the spatial and temporal trends of the low-cost sensor network to the method of calibration

We compared the differences in corrected $PM_{2.5}$ using similar methods to that in (Jin et al., 2019; deSouza et al., 2022) by calculating:





(1) The spatial root mean square difference (RMSD) between any two corrected exposures at
the same site: $SRMSD_{h,d} = \sqrt{\frac{1}{N}\sum_{i=1}^{N} (Conc_{hi} - Conc_{di})^2}$, where $Conc_{hi}$ and $Conc_{di}$ are
Jan 1- September 30, 2021 averaged PM$_{2.5}$ concentrations estimated from correction h and
d for site i. N is the total number of sites.
(2) The temporal RMSD between pairs of exposures: $TRMSD_{h,d} =$
$\sqrt{\frac{1}{M}\sum_{t=1}^{M} (Conc_{ht} - Conc_{dt})^2}$, where $Conc_{ht}$ and $Conc_{dt}$ are hourly corrected PM$_{2.5}$
concentrations averaged over all operational Love My Air sites estimated from correction h
and d for time t. M is the total number of hours of operation of the network.
(3) The spatial pearson correlation coefficient: $R_S =$
$\dfrac{\sum_{i=1}^{N} (Conc_{hi} - \underline{Conc_h})(Conc_{di} - \underline{Conc_d})}{\sqrt{\sum_{i=1}^{N} (Conc_{hi} - \underline{Conc_h})^2 \sum_{i=1}^{N} (Conc_{di} - \underline{Conc_d})^2}}$, where $\underline{Conc_h}$ and $\underline{Conc_d}$ are the average (across
all sites and times) corrected PM$_{2.5}$ concentrations estimated from corrections h and d
respectively.
(4) The temporal pearson correlation coefficient: $R_T =$
$\dfrac{\sum_{t=1}^{M} (Conc_{ht} - \underline{Conc_h})(Conc_{dt} - \underline{Conc_d})}{\sqrt{\sum_{t=1}^{M} (Conc_{ht} - \underline{Conc_h})^2 \sum_{i=1}^{N} (Conc_{dt} - \underline{Conc_d})^2}}$

We characterized the uncertainty in the 'corrected' PM$_{2.5}$ estimates at each site across the different
models using two metrics: a normalized range (NR) and uncertainty. NR for a given site represents
the spread of PM$_{2.5}$ across the different correction approaches.
(5) $NR = \frac{1}{M}\sum_{t=1}^{M} \dfrac{max_{k \in K}\ C_{kt} - min_{k \in K}\ C_{kt}}{\underline{C_t}}$
$C_{kt}$ is the PM$_{2.5}$ concentration at hour t from the kth model from the ensemble of K (which in this
case is 89) correction approaches. $\underline{C_t}$ represents the ensemble mean across the K different products
at hour t. M is the total number of hours in our sample for which we have PM$_{2.5}$ data for the site
under consideration.

For our sample (K = 89), we assume the variations in PM$_{2.5}$ across multiple models follows the t-
statistical distribution with the mean being the ensemble average. The confidence interval (CI) for
the ensemble mean at a given time t is:

(6) $CI_t = \underline{C_t} + t^* \dfrac{SD_t}{\sqrt{K}}$
Where $\underline{C_t}$ represents the ensemble mean at time t; t* is the upper (1-CI)/2 critical value for the t-
distribution with K-1 degrees of freedom. For K=89, t* for the 95% double tailed confidence
interval is 1.99. SD$_t$ is the sample standard deviation at time t.
(7) $SD_t = \sqrt{\dfrac{\sum_{k=1}^{K} (C_{k,t} - \underline{C_t})^2}{K-1}}$

We define an overall estimate of uncertainty as follows:
(8) $uncertainty = \frac{1}{M}\sum_{t=1}^{M} \quad t^* \frac{SD_t}{C_t\sqrt{K}}$ , which can also be expressed as
(8) $uncertainty = \frac{1}{M}\sum_{t=1}^{M} \quad \frac{\overline{CI_t - C_t}}{\underline{C_t}}$

## 3 Results

### 3.1 Evaluating the correction models at the co-location sites

When we evaluated each of the 21 correction models proposed on the entire co-location dataset
(**Tables 2** and **3**), we found that the C2 correction performed better overall than the C1, C3 and C4
corrections.

We also found that for corrections C3 and C4, more complex models yielded a better performance
(for example the RMSE for Model 16: 2.813 μg/m$^3$, RMSE for Model 2: 0.915 μg/m$^3$ generated
using the C3 correction) when evaluated during the period of co-location, alone (**Table S2**).
However, when models generated using the C3 and C4 correction were transferred to the entire
time period of co-location, we find that more complex multivariate regression models (Models 13-
16) and the machine learning model (Model 21) that include cos_time, performed significantly
worse than the simpler models. In some cases, these models performed worse even than the
uncorrected measurements. For example, applying Model 16 generated using C3 on the entire
dataset resulted in an RMSE of 32.951 μg/m$^3$ compared to 6.469 μg/m$^3$ for the uncorrected
measurements. Including data for another season in the training sample (C4), resulted in
significantly increased performance of the calibration over the entire dataset compared to C3,
although it did not result in an improvement in performance for all models compared to the
uncorrected measurements. For example, Model 16 generated using C4 yielded an RMSE of 6.746
μg/m$^3$. Among the multivariate regression models, we found that models of the same form that
corrected for RH instead of T or D did best. The best performance was observed for models that
included the nonlinear correction for RH (Model 12) or included an RH X T term (Model 5)
(**Tables 2** and **3**).

For corrections C1 and C2, we found that an increase in complexity of model form resulted in a
decreased RMSE. Overall, Model 21 yielded the best performance (RMSE = 1.281 μg/m$^3$ when
using the C2 correction, and 1.475 μg/m$^3$ when using the C1 correction with a LOSO CV and
1.480 μg/m$^3$ when using a LOBD correction). In comparison, the simplest model that corrected for
bias yielded an RMSE of 3.421 μg/m$^3$ for the C1 correction, and 3.008 μg/m$^3$ when using the C2
correction.

For correction C1, using a LOBD CV with the machine learning models resulted in better
performance than using a LOSO CV, except for Model 21 which is an RF model with additional
time-of-day and month covariates, for which performance using the LOSO was slightly better
(RMSE: 1.475 μg/m$^3$ versus 1.480 μg/m$^3$).

When we evaluated how well the models performed at high PM$_{2.5}$ concentrations (> 30 μg/m$^3$)
versus lower concentrations (≤ 30 μg/m$^3$), we found that multivariate regression models generated





489 using the C1 correction did not perform well in capturing spikes in PM$_{2.5}$ concentrations
490 (normalized RMSE > 25%). Multivariate regression models generated using the C2 correction
491 performed better (normalized RMSE ~ 20 -25 %). Machine learning algorithms generated using
492 both C1 and C2 corrections captured PM$_{2.5}$ spikes well (C1: normalized RMSE ~ 10 - 25%, C2:
493 normalized RMSE ~ 10 - 20%). Specifically, the C2 RF model (Model 21) yielded the lowest
494 RMSE values (4.180 μg/m$^3$, normalized RMSE: 9.8%), of all models considered. Machine learning
495 models generated using the C1 corrected that were tuned using LOBD CV instead of LOSO
496 performed better in both PM$_{2.5}$ concentration regimes. Models generated using C3 and C4 had the
497 worst performance in both concentration regimes and yielded poorer agreement with reference
498 measurements than even the uncorrected measurements. As in the case with the entire dataset,
499 more complex multivariate regression models and machine learning models generated using C3
500 and C4 performed worse than more simple models in both PM$_{2.5}$ concentration intervals (**Tables
501 S3** and **S4**).

503 We then evaluated how well the models generated using C1, C2, C3 and C4 corrections performed
504 when applied to minute-level LCS data at co-located sites. We found that the machine learning
505 models generated using C1 and C2 improved the performance of the LCS (Model 21 (CV=LOSO)
506 generated using C1 yielded an RMSE of 15.482 μg/m$^3$ compared to 16.409 μg/m$^3$ obtained from
507 the uncorrected measurements.) The more complex multivariate regression models yielded a
508 significantly worse performance across all corrections. (Model 16 generated using C1 yielded an
509 RMSE of 41.795 μg/m$^3$.) As in the case with the hourly-averaged measurements, using correction
510 C1, LOBD CV instead of LOSO for the machine learning models resulted in better model
511 performance except for Model 21. Few models generated using C3 and C4 resulted in improved
512 performance when applied to the minute-level measurements (**Tables 4** and **5**).

514 ***Table 4****: Performance of the calibration models developed using the co-located hourly*
515 *measurements to the minute-level data as captured using root mean square error (RMSE), and*
516 *Pearson correlation (R). LOSO CV was used to prevent overfitting in the machine learning models.*
517 *All corrected values were evaluated over the entire time period (April 23 - September 30, 2021).*

| ID | Name | Equation | C1 *Correction developed on data during the entire period of network operation* | | C2 *On-the-fly correction developed using data for the same week of measurement* | | C3 *Correction developed using measurements made in the first two weeks of January* | | C4 *Correction developed using measurements from the first two weeks of January and the first two weeks in May* | |
|---|---|---|---|---|---|---|---|---|---|---|
| | | | R | RMSE (μg/m$^3$) | R | RMSE (μg/m$^3$) | R | RMSE (μg/m$^3$) | R | RMSE (μg/m$^3$) |
| | **Raw Love My Air measurements** | | | | | | | | | |



| 0 | Raw | | 0.497 | 16.409 | - | - | - | - | - | - |
|---|---|---|---|---|---|---|---|---|---|---|
| | **Multivariate Regression (LOSO CV)** | | | | | | | | | |
| 1 | Linear | $PM_{2.5, corrected} = PM_{2.5} \times s1 + b$ | 0.497 | 15.667 | 0.498 | 15.646 | 0.497 | 15.657 | 0.497 | 15.663 |
| 2 | +RH | $PM_{2.5, corrected} = PM_{2.5} \times s_1 + RH \times s_2 + b$ | 0.495 | 15.678 | 0.500 | 15.618 | 0.492 | 15.721 | 0.494 | 15.686 |
| 3 | +T | $PM_{2.5, corrected} = PM_{2.5} \times s_1 + T \times s_2 + b$ | 0.496 | 15.670 | 0.500 | 15.621 | 0.493 | 15.822 | 0.495 | 15.671 |
| 4 | +D | $PM_{2.5, corrected} = PM_{2.5} \times s_1 + D \times s_2 + b$ | 0.497 | 15.663 | 0.498 | 15.640 | 0.491 | 15.805 | 0.495 | 15.693 |
| 5 | +RH x T | $PM_{2.5, corrected} = PM_{2.5} \times s_1 + RH \times s_2 + T \times s_3 + RH \times T \times s_4 + b$ | 0.499 | 15.634 | 0.500 | 15.621 | 0.495 | 15.669 | 0.498 | 15.640 |
| 6 | +RH x D | $PM_{2.5, corrected} = PM_{2.5} \times s_1 + RH \times s_2 + D \times s_3 + RH \times D \times s_4 + b$ | 0.496 | 15.671 | 0.500 | 15.622 | 0.477 | 15.892 | 0.494 | 15.684 |
| 7 | +D x T | $PM_{2.5, corrected} = PM_{2.5} \times s_1 + D \times s_2 + T \times s_3 + D \times T \times s_4 + b$ | 0.470 | 15.928 | 0.014 | 323.684 | 0.018 | 257.153 | 0.032 | 135.647 |
| 8 | +RH x T x D | $PM_{2.5, corrected} = PM_{2.5} \times s_1 + RH \times s_2 + T \times s_3 + D \times s_4 + RH \times T \times s_5 + RH \times D \times s_6 + T \times D \times s_7 + RH \times T \times D \times s_8 + b$ | 0.138 | 33.817 | 0.041 | 111.569 | 0.029 | 160.447 | 0.027 | 160.963 |
| 9 | PM x RH | $PM_{2.5, corrected} = PM_{2.5} \times s_1 + RH \times s_2 + RH \times PM_{2.5} \times s_3 + b$ | 0.494 | 15.688 | 0.501 | 15.615 | 0.485 | 15.896 | 0.486 | 15.844 |
| 10 | PM x D | $PM_{2.5, corrected} = PM_{2.5} \times s_1 + D \times s_2 + D \times PM_{2.5} \times s_3 + b$ | 0.498 | 15.644 | 0.499 | 15.630 | 0.477 | 16.145 | 0.491 | 15.820 |
| 11 | PM x T | $PM_{2.5, corrected} = PM_{2.5} \times s_1 + T \times s_2 + T \times PM_{2.5} \times s_3 + b$ | 0.495 | 15.675 | 0.501 | 15.610 | 0.483 | 17.172 | 0.495 | 15.675 |
| 12 | PM x nonlinear RH | $PM_{2.5, corrected} = PM_{2.5} \times s_1 + \frac{RH^2}{(1-RH)} \times s_2 + \frac{RH^2}{(1-RH)} \times PM_{2.5} \times s_3 + b$ | 0.496 | 15.659 | 0.497 | 15.650 | 0.494 | 15.705 | 0.495 | 15.681 |
| 13 | PM x RH x T | $PM_{2.5, corrected} = PM_{2.5} \times s_1 + RH \times s_2 + T \times s_3 + PM_{2.5} \times RH \times s_4 + PM_{2.5} \times T \times s_5 + RH \times T \times s_6 + PM_{2.5} \times RH \times T \times s_7 + b$ | 0.501 | 15.611 | 0.502 | 15.601 | 0.462 | 17.111 | 0.489 | 15.732 |
| 14 | PM x RH x D | $PM_{2.5, corrected} = PM_{2.5} \times s_1 + RH \times s_2 + D \times s_3 + PM_{2.5} \times RH \times s_4 + PM_{2.5} \times D \times s_5 + RH \times D \times s_6 + PM_{2.5} \times RH \times D \times s_7 + b$ | 0.496 | 15.657 | 0.502 | 15.602 | 0.460 | 17.710 | 0.479 | 15.948 |





| ID | Model | Equation | R | RMSE | R | RMSE | R | RMSE | R | RMSE |
|---|---|---|---|---|---|---|---|---|---|---|
| 15 | PM x T x D | $PM_{2.5, corrected} = PM_{2.5} \times s_1 + T \times s_2 + D \times s_3 + PM_{2.5} \times T \times s_4 + PM_{2.5} \times D \times s_5 + T \times D \times s_6 + PM_{2.5} \times T \times D \times s_7 + b$ | 0.134 | 35.196 | 0.020 | 217.684 | 0.012 | 178.589 | 0.044 | 114.530 |
| 16 | PM x RH x T x D | $PM_{2.5, corrected} = PM_{2.5} \times s_1 + RH \times s_2 + T \times s_3 + D \times s_4 + PM_{2.5} \times RH \times s_5 + PM_{2.5} \times T \times s_6 + T \times RH \times s_7 + PM_{2.5} \times D \times s_8 + D \times RH \times s_9 + D \times T \times s_{10} + PM_{2.5} \times RH \times T \times s_{11} + PM_{2.5} \times RH \times D \times s_{12} + PM_{2.5} \times D \times T \times s_{13} + D \times RH \times T \times s_{14} + PM_{2.5} \times RH \times T \times D \times s_{15} + b$ | 0.112 | 41.795 | 0.029 | 159.921 | 0.010 | 482.333 | 0.019 | 203.714 |
| | **Machine Learning (LOSO CV)** | | | | | | | | | |
| 17 | Random Forest | $PM_{2.5, corrected} = f(PM_{2.5}, T, RH)$ | 0.505 | 15.565 | 0.510 | 15.527 | 0.489 | 15.863 | 0.488 | 15.821 |
| 18 | Neural Network (One hidden layer) | $PM_{2.5, corrected} = f(PM_{2.5}, T, RH)$ | 0.496 | 15.669 | 0.501 | 15.611 | 0.495 | 15.699 | 0.477 | 16.202 |
| 19 | Gradient Boosting | $PM_{2.5, corrected} = f(PM_{2.5}, T, RH)$ | 0.500 | 15.625 | 0.502 | 15.604 | 0.485 | 15.779 | 0.486 | 15.765 |
| 20 | SuperLearner | $PM_{2.5, corrected} = f(PM_{2.5}, T, RH)$ | 0.500 | 15.622 | 0.503 | 15.591 | 0.483 | 15.805 | 0.490 | 15.719 |
| 21 | Random Forest | For C1: $PM_{2.5, corrected} = f(PM_{2.5}, T, RH, D, cos\_time, cos\_month, sin\_month)$ <br><br> For C2, C3, C4: $PM_{2.5, corrected} = f(PM_{2.5}, T, RH, D, cos\_time)$ | 0.514 | 15.482 | 0.512 | 15.502 | 0.481 | 16.349 | 0.481 | 16.185 |

518

519 **Table 5**: *Performance of the calibration models developed using the co-located hourly*
520 *measurements to the minute-level data as captured using root mean square error (RMSE), and*
521 *Pearson correlation (R). LOBD CV was used to prevent overfitting in the machine learning*
522 *models. All corrected values were evaluated over the entire time period (April 23 - September 30,*
523 *2021)*

| ID | Machine Learning (LOBD CV) | R | RMSE ($\mu g/m^3$) |
|---|---|---|---|



| 17 | Random Forest | $PM_{2.5, corrected} = f(PM_{2.5}, T, RH)$ | 0.506 | 15.561 |
|----|---------------|--------------------------------------------|-------|--------|
| 18 | Neural Network (One hidden layer) | $PM_{2.5, corrected} = f(PM_{2.5}, T, RH)$ | 0.496 | 15.666 |
| 19 | Gradient Boosting | $PM_{2.5, corrected} = f(PM_{2.5}, T, RH)$ | 0.501 | 15.610 |
| 20 | SuperLearner | $PM_{2.5, corrected} = f(PM_{2.5}, T, RH)$ | 0.503 | 15.594 (1.326) |
| 21 | Random Forest | $PM_{2.5, corrected} = f(PM_{2.5}, T, RH, D, cos\_time, cos\_month, sin\_month)$ | 0.510 | 15.516 |

## 3.1 Evaluating the representativeness of meteorological conditions at the co-location sites of the entire network

Temperature at the co-located sites across the entire period of the experiment during the development of C1 were similar to those at the rest of Love My Air network (**Figure S10**). The sensor CS19 is the only one that recorded lower temperatures than those at any of the other sites. Relative humidity at the co-located sites appears to be larger than at the other sites in the network (**Figure S11**).

We also compared meteorological conditions during the development of corrections C3 (Jan 1 - Jan 14, 2021) and C4 (Jan 1 - Jan 14, 2021 and May 1 - May 14, 2021), to those measured during the duration of network operation (C3: **Figures S12** and **S13**; C4: **Figures S14** and **S15**). Temperatures at the co-located sites during the development of C3 were on average lower than those reported during the operation of the network. Temperatures at the co-located sites during the development of C4 were more representative of the network than C3, although they too are smaller than the average temperatures experienced by the network. RH values during C3 and C4 tend to be on the higher side and are not representative of conditions experienced by some Love My Air sensors.

We then evaluated the transferability of the corrections developed.

## 3.2 Evaluating transferability at the co-location sites

**Figure 2** shows the performance (RMSE) of corrected Love My Air $PM_{2.5}$ data by generating corrections based on the 21 models previously proposed using the (a) C1 correction, CV= LOSO and CV = LODB for Models 17 - 21, when leaving out a test site (**Figure 2a**). Also shown is the result using the C1 correction when leaving out a three week period of data at a time and generating corrections based on the data from the remaining time periods across each site and using CV = LOBD for Models 17 - 21 (**Figure 2b**). Finally, **Figures 2c, 2d** and **2e illustrate** using the C2, C3 and C4 corrections, respectively, (CV= LOSO for Models 17 - 21) when leaving out a test site.

Large reductions in RMSE are observed when applying simple linear corrections (*Models 1 - 4*) to the uncorrected data across all corrections. Increasing the complexity of the model does not result





in marked changes in correction performance on different test sets for C1 and C2. Although the
performance of the corrected datasets did improve on average for some of the complex models
considered (*Model 17, 20, 21* for example, vis-a-vis simple linear regressions when using the C1
correction) (**Figures 2a, 2b**), this was not the case for *all* test datasets considered, as evinced by the
overlapping distributions of RMSE performances (e.g., Model 11 using the C2 correction resulted
in a worse fit for one of the test datasets). For C3 and C4, the performance of corrections was
worse across all datasets for the more complex multivariate model formulations (**Figures 2d**, **2e**),
indicating that using uncorrected data is better than using these corrections and calibration models.
Wilcoxon tests and t-tests (based on whether Shapiro-Wilk tests revealed that the distribution of
RMSEs was normal) revealed significant improvements in the distribution of RMSEs for all
corrected test sets vis-a-vis the uncorrected data. There was no significant difference in the
distribution of RMSE values from applying C1 and C2 corrections to the test sets, across the
different models. For corrections C3 and C4, we found significant differences in the distribution of
RMSEs obtained from running different models on the data, implying that the choice of model has
a significant impact on transferability of the calibration models to other monitors.





Figure 2: *Performance (RMSE) of corrected Love My Air PM$_{2.5}$ data by generating corrections based on the 21 models previously proposed using **(a)** Correction C1 when leaving out a co-location site in turn and then running the generated correction on the test site (Note that for machine learning models (Models 17- 21), we performed CV using a LOSO CV as well as a LOBD CV approach), **(b)** Correction C1 when leaving out 3 week periods of data at a time and generating corrections based on the data from the remaining time periods across each site, and evaluating the performance of the developed corrections on the held out 3 weeks of data (Note that for machine learning models (Models 17- 21), we performed CV using a LOBD CV approach), **(c)** Correction C2 when leaving out a co-location site in turn and then running the generated correction on the test site, **(c)** Correction C3 when leaving out a co-location site in turn and then running the generated correction on the test site, **(c)** Correction C4 when leaving out a co-location site in turn and then running the generated correction on the test site. Each point represents the*





*RMSE for each test dataset permutation. The distribution of RMSEs is displayed using boxplots*
*and violinplots*

The time-series of corrected $PM_{2.5}$ values for Models 1, 2, 5, 16, and 21 (RF using additional
variables) (using CV = LOSO for the machine learning Models 17 and 21) for corrections
generated using C1, C2, C3 and C4 are displayed in **Figure 3** for Love My Air sensor CS 1. These
subsets of models were chosen as they cover the range of model forms considered in this analysis.

From **Figure 3**, we note that although the different corrected values from C1 and C2 track each
other well, there are small systematic differences between the different corrections. Peaks in
corrected values using on-the-fly data tend to be higher than those using archived data. Peaks in
corrected values using machine learning methods on the archived data are higher than those
generated from multivariate regression models. There are marked differences in the corrected
values from C3 and C4. Specifically Model 16 yields peaks in the data that corrections using the
other models do not generate. This pattern was consistent when applying this suite of corrections to
other Love My Air sensors.

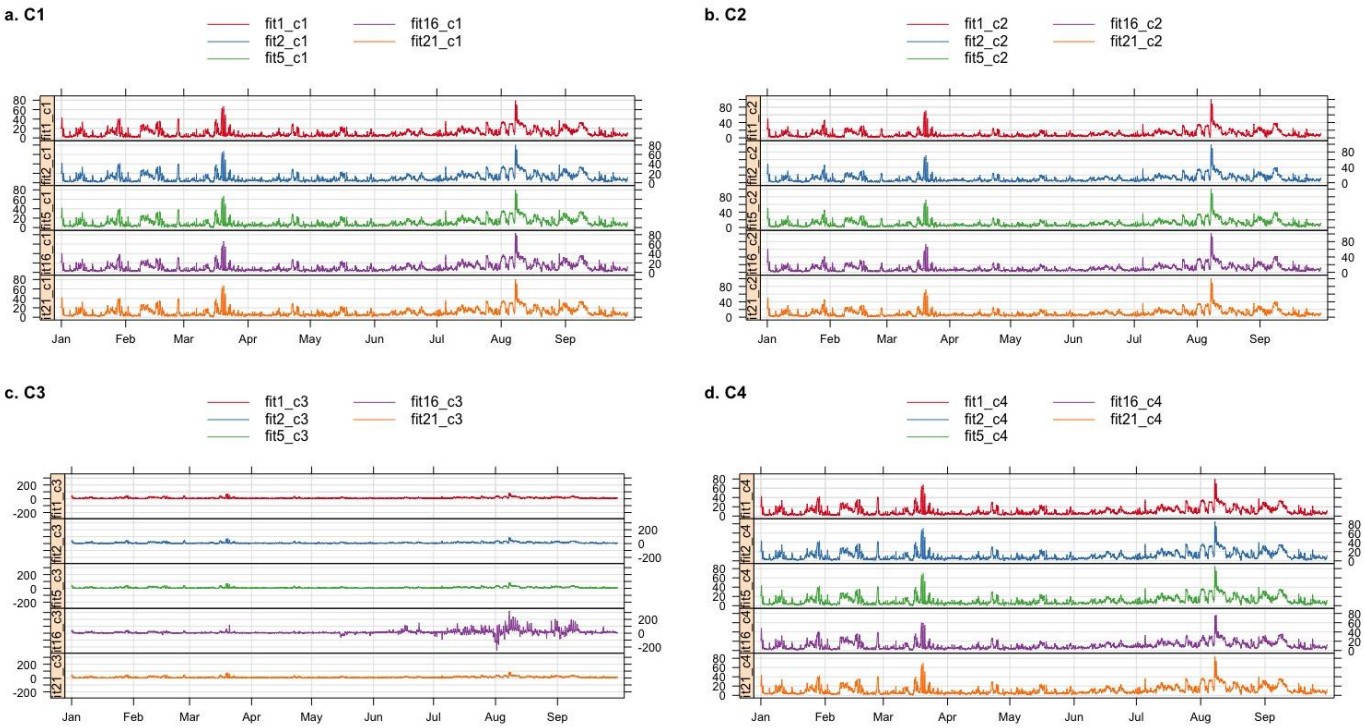

***Figure 3***: *Time-series of the different $PM_{2.5}$ corrected values for Models 1, 2, 5, 16 and 21 across*
*corrections (a) C1, (b) C2, (c )C3 and (d) C4  for the Love My Air monitor CS1*



## 3.3 Evaluating the sensitivity of hotspot detection across the network of sensors to the calibration method


Mean (95% CI) PM$_{2.5}$ concentrations across the different models (as well as CV technique) and
corrections (26 (C1) + 21 x 3 (C2, C3, C4) = 89 listed in **Tables 2** and **3**) at each Love My Air site
for the duration of the experiment (Jan 1 - September 30, 2021) are displayed in **Figure S16**. Due
to overlap between the different corrected measurements across sites, identification of the most
polluted site is dependent on the correction algorithm used. We examined the sensitivity of the
'most polluted site' at different time-intervals
Every hour, we ranked the different monitors for each of the 89 different corrections. We found
that there were on average 4.4 (median = 5) monitors that were ranked most polluted. When this
calculation was repeated using daily-averaged corrected data, there were on average 2.5 (median =
2) monitors that were ranked the most polluted. The corresponding value for weekly-corrected data
was 2.4 (median = 1), and for monthly data was 3 (median = 3) (**Figure 4**).

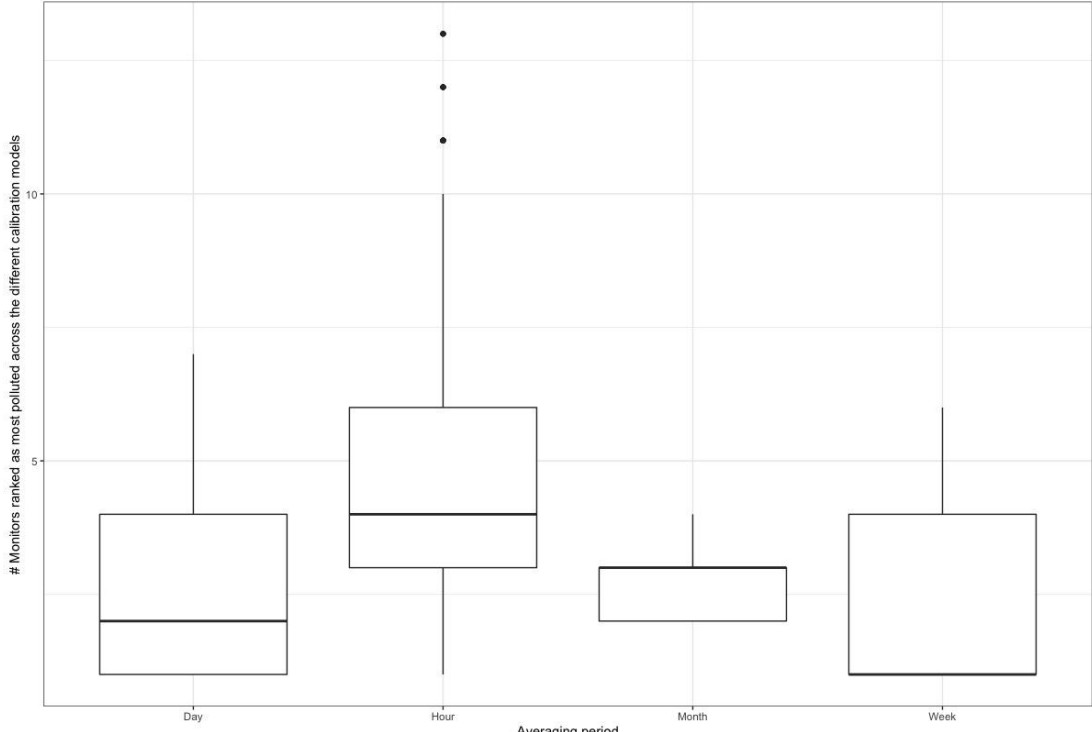

***Figure 4:*** *Variation in the number of sites that were ranked as 'most polluted' across the 89*
*different corrections for different time-averaging periods displayed using boxplots*

## 3.4 Evaluating sensitivity of the spatial and temporal trends of the low-cost sensor network to the method of calibration


The spatial and temporal RMSD values between corrected values generated from applying each of
the 89 models using the four different correction approaches across all monitoring sites in the Love
My Air network are displayed **Figures 5** and **6**, respectively. It appears that there is larger temporal





variation (max 32.79 μg/m³), in comparison to spatial variations displayed across corrections (max:
11.95 μg/m³). Model 16 generated using the C3 correction has the greatest spatial and temporal
RMSD in comparison with all other models. Models generated using the C3 and C4 corrections
displayed the greatest spatial and temporal RMSD vis-a-vis C1 and C2. **Figures S17- S20** display
spatial RMSD values between all models corresponding to corrections C1-C4, respectively.
**Figures S21- S24** display temporal RMSD values between all models corresponding to corrections
C1-C4, respectively. Across all corrections the temporal RMSD between models is greater than the
spatial RMSD.
Spatial and temporal correlation coefficients between corrected measurements generated from
applying all 89 models using the four different correction approaches across the entire network are
displayed in **Figures S25** and **S26**, respectively. The spatial correlations are lower than temporal
correlations between corrected measurements.



**Figure 5**: *Spatial RMSD (μg/m³) calculated using the method detailed in section 2.3.4 from applying each of the 89 models using the four different correction approaches to all monitoring sites in the Love My Air network*





*Figure 6*: *Temporal RMSD (μg/m³) calculated using the method detailed in section 2.3.4 from applying each of the 89 models using the four different correction approaches to all monitoring sites in the Love My Air network*

The distribution of uncertainty and the NR in hourly corrected measurements over the 89 models by monitor are displayed in **Figure 7**. Overall, there are small differences in uncertainties and NR of the exposure assessment across sites. The average NR and uncertainty across all sites are 1.554





653 (median: 0.9768) and 0.044 (median: 0.033) , respectively.  We note that although the uncertainties
654 in the data are small, the average normalized range tends to be quite large.

655

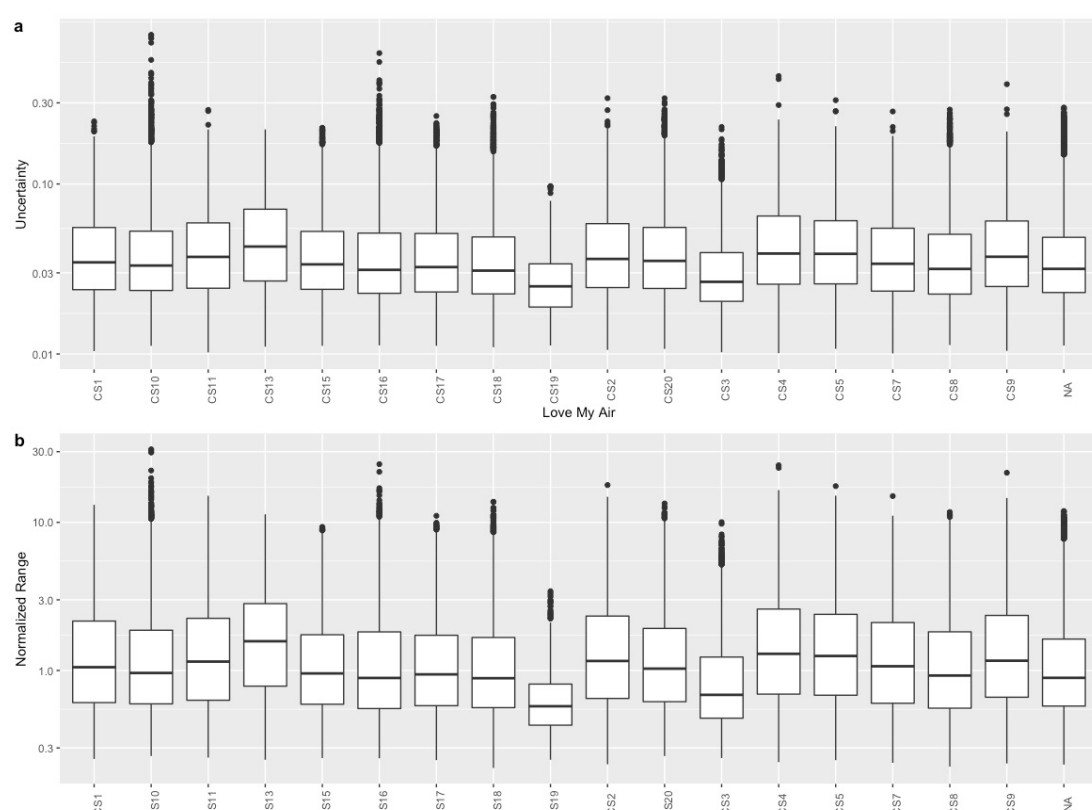

656

657 ***Figure 7**: Distribution of (a) uncertainty and (b) normalized range (NR) in hourly-corrected*
658 *measurements across all 89 correction models at each site using the methodology described in*
659 *Section 2.3.4*

## 4 Discussion and Conclusions

661 In our analysis of how transferable the correction algorithms developed at the Love My Air co-
662 location sites are to the rest of the network, we found that for C1 and C2 corrections, more
663 complex models yielded a better fit at the co-location sites. When examining the C3 and C4
664 corrections, we found that although these corrections appeared to significantly improve LCS
665 measurements for the time period of model development (**Table S2**), when applied to the entire
666 time period of operation they did not perform well. Many of the models, especially the more
667 complex multivariate regression models, performed significantly worse than even the uncorrected
668 measurements. This indicates that calibration models generated during short-term time periods,
669 even if the time periods correspond to different seasons, may not necessarily transfer well to other
670 times, likely due to changes in the aerosol composition, and differences in meteorological
671 conditions, among other potential factors. This suggests the need for calibration models to be
672 developed over longer time periods that better capture different LCS operating conditions. For C3


and C4, we found models that relied on nonlinear formulations of RH, that serve as proxies for
hygroscopic growth, yielded the best performance, as compared to more complex models. This
suggests that physics-based calibrations are potentially an alternative approach when relying on
short co-location periods and need to be explored further.

When evaluating how transferable the calibration models using the different correction approaches
were to the rest of the network, we found that for C1 and C2, more complex models that appeared
to perform well at the co-location sites did not necessarily transfer best to the rest of the network.
Specifically, when we tested these models on a co-located site that was left out when generating
the correction, we found that some of the more complex models run using the C2 correction
yielded a significantly worse performance at some test sites (**Figure 2**). If the corrected data were
going to be used to make site-specific decisions then such corrections would lead to important
errors. When evaluating C3 and C4 correction approaches we observed a large distribution of
RMSE values across sites. For several of the more complex models developed using C3 and C4
corrections, the RMSE values were larger than observed for the uncorrected data, suggesting that
certain calibration models could result in even more error-prone data than using uncorrected
measurements.

For C1 and C2, we found that there were no significant differences in the distribution of the
performance metric: RMSE of corrected measurements from simpler models in comparison to
those derived from more complex corrections at test sites (**Figure 2**). For C3 and C4, we found
significant differences in the distribution of RMSE across test sites, which indicates that these
models are likely site-specific and not easily transferable to other sites in the network. This
suggests that less complex models might be preferred when short-term co-locations are carried out
for sensor calibration.

Our findings reinforce the idea that evaluating calibration models at all co-location sites using
overall metrics like RMSE should not be seen as the only/best way to determine how to calibrate a
network of LCS. Instead, approaches like LOSO, LOBD, or a combination of these, as
demonstrated should be used to evaluate calibration transferability.

We also found that the calibration models yielded different performance results at different $PM_{2.5}$
concentration ranges. Machine learning models developed using C1, and models developed using
C2 were better than multivariate regression models generated using C1 at capturing peaks in
pollution (> 30 $\mu g/m^3$). All models using C3 and C4 yielded poor performance results across both
concentration ranges ($PM_{2.5} > 30$ $\mu g/m^3$ and $PM_{2.5} \leq 30$ $\mu g/m^3$).

When evaluating how well the calibration models translated to minute-level data (**Tables 4** and **5**),
we observed that machine learning models generated using C1 and C2, improved the LCS
measurements. More complex multivariate regression models performed poorly. All C3 and C4
models also performed poorly. This suggests that caution needs to be exercised when transferring
models developed at a particular time scale to another (**Tables S3** and **S4**).



Our findings thus far indicate that different calibration approaches are required for different end
purposes. There may not be a single one-size-fits-all calibration approach.

We found that the 'most polluted' site in the Love My Air network was dependent on the
calibration algorithm used on the network. We found that for the Love My Air network, the
detection of the most polluted site was sensitive to the duration of time-averaging of the corrected
measurements (**Figure 4**). Hotspot detection was most robust using weekly-averaged
measurements. Such an analysis thus reveals the most robust temporal scale for decision-making
related to evaluating hotspots.

We found that the temporal RMSD (**Figure 6**) was greater than the spatial RMSD (**Figure 5**) for
the ensemble of 47 corrected exposure assessments developed for the Love My Air network. One
of the reasons this may be the case is that $PM_{2.5}$ concentrations across the different Love My Air
sites in Denver are highly correlated (**Figure S5**), indicating that the contribution of local sources
to $PM_{2.5}$ concentrations in Denver is small. Due to the low variability in $PM_{2.5}$ concentrations
across sites, it makes sense that the variations in the corrected $PM_{2.5}$ concentrations will be seen in
time rather than space. The largest pairwise temporal RMSD were all seen between corrections
derived from complex models using the C3 correction.

However, we note that the temporal correlation coefficients (**Figure S26**) for all-pairwise
correction models were higher than the corresponding spatial coefficients (**Figure S25**). This
implies that although the corrections generated from all models considered tended to track each
other (except for a few models using C3) some corrected values were biased low, whereas some
were biased high. It's important to understand under what conditions these biases occur. One of the
ways this can be determined is by evaluating the performance of the calibrated data under different
conditions, such as in different pollution regimes as demonstrated in this paper (**Tables S3** and **S4**).

Finally, we observed that the uncertainty in $PM_{2.5}$ concentrations across the ensemble of
corrections was consistently small for the Love My Air Denver network. The normalized range in
the corrected measurements, on the other hand, was large, indicating that the corrections yield a
large range of corrected measurements; however, most of the corrected measurements fall within a
relatively small interval. Thus, deciding which calibration algorithm to pick has important
consequences for decision-makers using data from this network.

In summary: this paper makes the case that it is not enough to evaluate calibration algorithms
based on metrics of performance at co-located sites, alone. We need to:

1) Evaluate models under different conditions (e.g., pollution concentrations) to evaluate the
circumstances under which different calibration algorithms do well to determine which model to
use for which use-case.



2) Determine how well calibration adjustments can be transferred to other locations. Specifically,
although we found that in Denver some corrections performed well at co-location sites, they could
result in large errors at specific sites that would create difficulties for site-specific decision making.
3) Examine how well calibration adjustments can be transferred to other time periods. In this study
we found that models developed using the C3 correction were not transferable to other time
periods because the conditions during the co-location were not representative of broader operating
conditions in the network.
4) Evaluate how well calibration algorithms developed for a specific time-scale transfer to
measurements at other time intervals.
5) Use a variety of approaches to quantify transferability, both focusing on co-location sites (using
a LOSO and/or LOBD cross-validation scheme) and looking at the wider low-cost sensor network
(e.g., with spatio-temporal correlations and RMSD). The metrics proposed in this paper to evaluate
model transferability can be used in other networks.
6) Investigate how adopting a certain timescale for averaging measurements could mitigate the
uncertainty induced by the calibration process. Namely, we found that in the Love My Air
network, hotspot identification was more robust to using daily-averaged data than hourly-averaged
data.
In this work, the Love My Air network under consideration is located over a fairly small area in a
single city. In this network, for the time period considered, $PM_{2.5}$ seems to be mainly a regional
pollutant and the contribution of local sources is small. More work needs to be done to evaluate
model transferability in networks in other settings. Concerns about model transferability are likely
to be even more key when thinking about larger networks that span different cities and should be
considered in future research.

## Author Contributions

PD conceptualized the study, developed the methodology, carried out the analysis and wrote the first draft.
TS and WO provided PD with access to the data. PD and BC obtained funding for this study. BC produced
Figure 1. All authors helped in refining the methodology and editing the draft.

## Acknowledgements

PD and BC gratefully acknowledge a CU Denver Presidential Initiative grant that supported their
work. The work of R. Kahn is supported in part by NASA's Climate and Radiation Research and
Analysis Program under Hal Maring, as well as NASA's Atmospheric Composition Program under
Richard Eckman. The authors are grateful to the Love My Air team for setting up and maintaining
the Love My Air network. The authors are also grateful to Carl Malings for useful comments.

## Data Availability

The data used in this study can be obtained from the author on request



## Competing Interests

The authors declare that they have no conflict of interest.

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
