# Peer review of "Calibrating Networks of Low-Cost Air Quality Sensors"

_Atmospheric Measurement Techniques, 2022_

## Author Comment (AC2)

Reviewer 1:

This is a very timely paper that provides a systematic and deep analysis on the different ways that low cost sensors can be calibrated by colocation with regulatory grade equipment. In particular, it provides useful information on how best calibrate depending on the colocation period possible. The paper uses a variety of calibration models (n=21) starting with simple linear corrections, and ending with complex machine learning algorithms, where it is often difficult to know the mechanism of the correction. The calibration models are tested on four different colocation periods. In particular the difference between the C1 and C2 colocation strategies is interesting because it shows that more calibration data is not necessarily helpful if it doesn't capture the variability in the parameters. The hot spot analysis is also interesting, highlighting the need for care when interpreting individual sensors within a network.

Low cost sensors are used in various ways. Sensor networks like the 'love my air' network used as the data set in this paper are used to complement existing regulatory activities, whereas in other contexts low cost sensors are used where regulatory measurements are scant or non-existent. This paper will provide very useful to all users of low cost sensors.

The paper is very robust in its description and should be published, once the following (mostly minor) points are addressed.

> Authors: Many thanks for this assessment of our work

In general, the resolution of the figures should be improved.

> Authors: Thank you. We have improved their resolution in this revision.

Abstact and L49 – no need to say 'gold standard reference monitors', 'reference monitors' is sufficient.

> Authors: Thank you. We have removed the extraneous term

L42 estimates vary widely for number of premature deaths due to air pollution, this should be acknowledged, or at least the prefix of 'approximately' should be added by the 6.7M.

> Authors: Thank you. We have included the pre-fix 'approximately'

L70 'leading to mass overestimation…' should be 'leading to the (regulatory) dry mass overestimation' or similar

> Authors: Thank you. We have noted that we mean dry mass estimation

L74 need to acknowledge that most of the PM mass concentration is at particle diameters greater than 300 nm.

> Authors: Thank you we have done so

> "LCS are not able to detect particles with diameters below a specific size, which is determined by the wavelength of laser light within each device, and is generally in the vicinity of 0.3 μm, whereas the peak in pollution particle number size distribution is typically smaller than 0.3 μm."

L96 Köhler not kohler

> Authors: Thank you. We have made this change

L119 I would state that R^2 is a misleading indicator rather than might be

> Authors: Thank you. We have made the suggested change

L215-216 you would expect averaged data to have less variance.

> Authors: The necessary scale of the plots (to capture spikes of minute-level PM$_{2.5}$ as high as ~1000 μg/m$^3$) which perhaps make it harder to evaluate variability. When we zoom into a smaller subset of sensors as in Figures S4 we see a high degree of variability

L240 RH, T, and D are not independent parameters. A discussion of the use of non-independent parameters within the calibration algorithms should be provided.

> Authors: The reviewer is quite right. We note, the following when describing D:
>
> "We derived dew-point (D) from T and RH reported by the Love My Air sensors using the *weathermetrics* package in the programming language R (Anderson and Peng, 2012), as D has been shown to be a good proxy of particle hygroscopic growth in previous research (Barkjohn et al., 2021; Clements et al., 2017; Malings et al., 2020). Some previous work has also used a nonlinear correction for RH in the form of RH$^2$/(1-RH), that we also calculated for this study."
>
> We note that we use D, in addition to T and RH because for all our multilinear regressions we used the same set of equations that US EPA researchers used when deriving a national calibration equation for PurpleAir monitors.
>
> To whit in our note on statistical modeling we add the following note:
>
> "Sixteen models were multivariate models that were used in a recent paper (Barkjohn et al., 2021) to calibrate another network of low-cost sensors: the PurpleAir, that rely on the same PM$_{2.5}$ sensor (Plantower) as the Canary-S monitors in this study. As T, RH and D are not independent (**Figure S8**), the 16 linear regression models include adding the meteorological conditions considered as interaction terms, instead of additive terms. The remaining 5 relied on machine learning techniques."

L302 how do you choose which site to leave out in the LOSO methodology?  What potential bias(es) does this introduce into the analysis?

> Authors: Thank you for this question. We left out each site in turn and used models developed for the other sites to make predictions at the left out site. We chose the model that yielded the best average performance across each of the left out sites. We include this description in the text:

> "We used a Leave-One-Site (I25 Globeville, I25 Denver, La Casa, CAMP)-Out (LOSO) approach for cross validation (CV) to prevent overfitting in our machine learning models (Models 17 - 21 in **Table 2**). Briefly, we split the data into four groups, with each group excluding data from a single reference monitoring site. In each cross-validation iteration, we selected each group in turn to fit the model and made predictions at the left-out site. **The model that had the best average performance across all the left out sites was chosen. In this manner this CV approach was used to tune the hyper parameters in the machine learning models adopted in this study using correction approaches: C1, C2, C3 and C4.**"

L333 and most other equations. Pet peeve – use proper multiply symbol rather than x in equations.

> Authors: Thanks. We have made this change everywhere.

L351 "as these concentrations account for the greatest differences in health and air pollution avoidance behavior impacts" this statement is unclear. Are you suggesting that 30 ug/m3 is a cut off for more harmful PM health effects?  My understanding is the health effect: concentration curve is reasonably linear over these ranges.

> Authors: Thank you for this note. Our choice of this threshold is derived from the way in which AQ Health Index (AQHI) is derived. More information on this threshold can be found in the paper we cited: Nilson et al., (2022)

L393 note a p value of 0.05 means that 1/20 results can be to chance.  With 21 models and 4 colocation conditions, you might expect some false positives.

> Authors: Thanks for this note. In the context of this paper, we compared the distribution of errors in prediction on each test dataset when leaving out a given site to errors derived from using data at all co-location sites. A $p < 0.05$, indicates that for each comparison, there is a 95% probability that the errors belong to the same distribution. This threshold is widely used.

L457 model 2 has a lower RMSE than model 16, so doesn't that contradict "more complex models yielded a better performance"

> Authors: Thank you for catching this mistake. For Model 2, instead of listing the RMSE, we made a mistake and listed R instead. We have corrected this as follows:

"We also found that for corrections C3 and C4, more complex models yielded a better performance (for example the RMSE for Model 16: 2.813 µg/m$^3$, **RMSE for Model 2: 3.110 µg/m$^3$** generated using the C3 correction) when evaluated during the period of co-location, alone (**Tables S2** and **S3**)."

L472 "the nonlinear correction for RH" gave best performance. Doesn't this suggest a model using a physically reasonable model (essentially k-Köhler) works best when extensive colocation data is not possible. See for example Crilley et al. (2020) https://doi.org/10.5194/amt-13-1181-2020

Authors: Thanks! We indeed make this observation in the Discussion and think it is an important take-away from this paper:

"For C3 and C4, we found models that relied on nonlinear formulations of RH, that serve as proxies for hygroscopic growth, yielded the best performance, as compared to more complex models. This suggests that physics-based calibrations are potentially an alternative approach when relying on short co-location periods and need to be explored further."

L528 does the temperature offset on CS19 make sense with respect to the position of the sensor?

Authors: Yes it does. It is in the shade.

Reviewer 2:

General comments

This paper is about calibrating low-cost sensors of particulate matter using many different models. The paper promises to give a set of best practices and to describe the transferability of the calibration to sensors not co-located with a reference measurement; however, there is so much data in the paper that these tangible conclusions are lost to me. Maybe some of my comments below will help bring clarity to the next version of this paper.

Authors: Thanks for this note. We have significantly restructured the paper to highlight our key contributions and to make it easier to read.

We have updated the Abstract to read as follows to make our intent in the paper clearer:

"Ambient fine particulate matter (PM$_{2.5}$) pollution is a major health risk. Networks of low-cost sensors (LCS) are increasingly being used to understand local-scale air pollution variation. However, measurements from LCS have uncertainties that can act as a potential barrier for effective decision-making. LCS data thus need adequate calibration to obtain good quality PM$_{2.5}$ estimates. **In order to develop calibration factors, one or more LCS are typically co-located with reference monitors for short- or long-periods of time.** A calibration model is then developed that characterizes the

relationships between the raw output of the LCS and measurements from the reference monitors. This calibration model is then typically *transferred* from the co-located sensors to other sensors in the network. Calibration models tend to be evaluated based on their performance only at co-location sites. **It is often implicitly assumed that the conditions at the relatively sparse co-location sites are representative of the LCS network overall, and that the calibration model developed is not overfitted to the co-location sites. Little work has explicitly evaluated how transferable calibration models developed at co-location sites are to the rest of an LCS network, even after appropriate cross-validation. Further, few studies have evaluated the sensitivity of key LCS use-cases such as hotspot detection to the calibration model applied. Finally, there has been a dearth of research on how the duration of co-location (short-term/long-term) can impact these results.** This paper attempts to fill these gaps using data from a dense network of LCS monitors in Denver deployed through the city's Love My Air program. **It offers a series of transferability metrics for calibration models that can be used in other LCS networks and some suggestions as to which calibration model would be most useful for achieving different end goals."**

We have also made substantial changes to the Introduction and Methods section to make the goal of this paper clear. For example. Section 2.3 contains an outline of the key questions we ask in this paper:

"**Uncorrected Love My Air measurements tend to be biased upwards from the corresponding reference PM$_{2.5}$ levels by an average of ~12% (Figure S9). We first evaluate:**
1) **Were meteorological conditions at the co-location sites representative of network operating conditions?**
2) **How well do different calibration models perform when using the traditional method of model evaluation at co-location sites, during the period of co-location?**

**We then evaluate transferability of the calibration models in time and space by evaluating:**
1) **How well do calibration models developed during short-term co-locations (corrections: C3 and C4) perform when transferred to long-term network measurements?**
2) **How well do calibration models developed at a small number of co-locations sites transfer in space to other sites, even after appropriate cross-validation to prevent overfitting?**
3) **Different metrics to quantify the uncertainty in spatial and temporal trends in PM$_{2.5}$ reported by the LCS network to the calibration model applied.**

**Finally, we evaluate the impact of the choice of calibration model on key LCS network use-cases, such as hotspot detection, or detection of the most-polluted**

**site. In supplementary analyses, we also evaluate how much the calibration model impacts the following additional use-cases:**

1) **LCS are increasingly used to evaluate pollution trends on increasingly short timescales. We evaluated how well calibration models developed using hourly aggregated data to minute-level LCS measurements**
2) **LCS have been deployed to track smoke from fires. We evaluate how well different calibration models perform at high PM$_{2.5}$ concentrations."**

Also, in the conclusions, future work #2 is exactly what this paper was supposed to determine (based on what the abstract tells us). Thus, there may be a big problem with the overall scope of this paper and confusion over exactly what the take-home messages should be from this work.

Authors: Thank you for this comment. We have substantially rewritten this paper to define our objectives more clearly. The numbered list at the end of the discussion and conclusion section is intended to highlight the implications of our work for other low-cost sensor studies. Namely, we list steps that other managers of LCS networks need to carry out to ensure that the calibration model derived is transferable across space and time in the network and is sufficient for the end-use of the data.

For better readability of the final paper, consider breaking up the big tables into a few smaller ones with more focused information in them. It might be worth using color or shading to indicate the sensors or models that stand out and are talked about more in the text.

Authors: Thank you for this comment. We have taken the reviewers advice and have majorly restructured the paper and

There are so many references to Supplemental figures, do some of these perhaps belong in the main paper? Maybe display data for a specific site or model and then have the rest of the sites and models in the Supplemental. But then, the reader gets more out of the main paper as a standalone manuscript.

Authors: Some analyses that were not central have been moved to the SI. We now have minimal references to the SI

Lines 333 and 453 - Which is it, 89 models or 21 models?

Authors: There are 89 models across corrections C1, C2, C3 and C4. For each correction we used a set of 21 calibration algorithm forms listed in Table 1. For the five machine learning algorithm forms, for correction C1, we also evaluated if CV=LOBD instead of LOSO impacted the results. C2, C3 and C4 are short-term corrections for which we only used CV = LOSO for the machine learning models. In order to make this more explicit we updated the text as follows:

> "Overall, we test 89 calibration models (21 (C1, CV=LOSO) + 5 (C1, CV=LOBD) + 21 × 3 (C2, C3, C4) = 89)"

Is Section 2.3.1 (and really, all of Section 2.3) necessary?  The way this section is presented, I'm not sure what value it adds to the paper (except for the equations).  There are lots of statements about 'we report' and 'we display', but doesn't say where to find these.

> Authors: Thanks. We have completely revised our Methods and Results section based on this comment.

Line 454 seems to be an important conclusion, but I don't see any good defense of this statement in the rest of the paper.  How is the C2 correction better exactly?  In fact, line 683 says that the C2 correction was significantly worse for the complex models (were complex and simple models clearly defined anywhere?).  Lines 567 says that C1 and C2 corrections have no significant differences between them.  These statements seem like contradictions to me and help lead to my confusion about the whole paper.

> Authors: Thank you. We have completely revised the Conclusions to make it more clear. For example, this paragraph now reads:
>
> "When we evaluated each of the 21 correction models proposed on the entire co-location dataset, we found that based on R and RMSE values the on-the-fly C2 correction performed better overall than the C1, C3 and C4 corrections for most calibration model forms (**Tables 2** and **3**).
>
> Within corrections C1 and C2, we found that an increase in complexity of model form resulted in a decreased RMSE. Overall, Model 21 yielded the best performance (RMSE = 1.281 μg/m$^3$ when using the C2 correction, and 1.475 μg/m$^3$ when using the C1 correction with a LOSO CV and 1.480 μg/m$^3$ when using a LOBD correction). In comparison, the simplest model that corrected for bias yielded an RMSE of 3.421 μg/m$^3$ for the C1 correction, and 3.008 μg/m$^3$ when using the C2 correction."

Line 670 - The statement here is not very certain; it seems to say that differences in meteorology "likely" matter.  Can't this paper quantify the influence of meteorology?  You have T and RH data at each sensor, so you should be able to better determine the effects of meteorology as compared to aerosol composition, where you have no measurements to use.

> Authors: Thanks for this comment. We hope to do so in future work. At the moment, the paper merely demonstrates that we can't take transferability of calibration algorithms as a given and devises metrics that users can use to evaluate transferability. In future work we will do a deep dive into how factors such as meteorology affect transferability.

Line 704-705 If this is an important conclusion, then there should a figure in the main paper that supports this conclusion (I don't think there is).

> Authors: We decided that this analysis was not central to the paper and have moved this to the supplement

Stylistically, much of the Discussion (Section 4) doesn't seem to add anything new; it's just repeating the conclusions from each of the figures presented earlier.

> Authors: In Section 4, we try and discuss key results. For example, we discuss the importance of exploring physics-based calibrations when short-term co-locations are carried out. We discuss what our results mean for researchers planning on deploying networks of low-cost sensors.

Line 730 - Can you really conclude this about Denver? Later, Line 779, you state that the network was over a "fairly small area". Was all of Denver covered, then?

> Authors: We provided a map of LoveMyAir sites with a distance scale in Figure 1. We have revised this sentence to read:
>
> "We found that the temporal RMSD (**Figure 7**) was greater than the spatial RMSD (**Figure 6**) for the ensemble of 47 corrected exposure assessments developed for the Love My Air network. One of the reasons this may be the case is that $PM_{2.5}$ concentrations across the different Love My Air sites in Denver are highly correlated (**Figure S5**), indicating that the contribution of local sources to $PM_{2.5}$ concentrations in the Denver neighborhoods in which Love My Air was deployed is small. Due to the low variability in $PM_{2.5}$ concentrations across sites, it makes sense that the variations in the corrected $PM_{2.5}$ concentrations will be seen in time rather than space. The largest pairwise temporal RMSD were all seen between corrections derived from complex models using the C3 correction."

Line 741 - Did you define or identify different pollution regimes somewhere?

> Authors: This is defined in section 2.3.7. Hopefully, the rewording of the Methods section will make this easier to find.

There are a number of sentences throughout the paper which use "it" or "this" as the subject in the sentence, which can add confusion and ambiguity to those sentences. Consider rewording all of these instances.

> Authors: Thank you. We have gone through the document and tried to delete as many of these instances as possible.

 Specific comments

Line 42 - Can you find a more recent citation and statistic? References says you last accessed the website almost 2 years ago.

> Authors: Thank you. This is the most recent statistic from the SOGA website.

Line 125-127 - What about sensor-to-sensor variability?  Why is that not considered?  Are these sensors all cross-calibrated in a lab prior to deployment?

Authors: There is little sensor to sensor variability. Three sensors were co-located at the I25 Globeville site and show high correlation as indicated in section 2.1.2

Line 201 - Is this correlation for minute or hr time resolution?

Authors: hourly data. We noted that unless specific we were referring to hourly data everywhere in the text. We have revised section 2.1.2 to make this more clear.

Line 217 - What additional uncertainties?  Be specific.

Authors: Thanks we have made the following change:

"As can be seen, the data at the minute-level displays more variation and peaks in $PM_{2.5}$ concentrations than the hourly-averaged measurements (**Figure S7)**, likely due to the impact of passing sources. It is also important to mention that minute-level reference data may have some **additional uncertainties introduced due to instrument error** given the finer time resolution. Unless explicitly referenced, we will be reporting results from using hourly-averaged measurements."

Line 330-331 - Figure S5 and S6 don't actually prove that there is a high correlation across sites.

Authors: Figure S5 shows correlations across each site. Figure S6 displays the time-series of each of the co-located sensor at I25-Globeville and shows that the pattern is very similar

Line 335 - Figure S9 seems pretty important to some conclusions stated later; I wonder if this should be in the main paper?  Also, does this include all of the co-located sites or just some of them?  (I think it must be all the reference monitor sites but just one of the LCS at those sites even though there may be multiple.). Be specific in the text and figure caption.  Also, the colorbar is missing labels.

Authors: We have made sure that the scale in Figure S9 is explicitly labelled. We have moved all relevant figures to the main text. We have adjusted the caption to appropriately say that this scatterplot has been produced for all co-located measurements

Lines 503-505 - I am confused about what the 1-minute data are being compared to to evaluate the LCS performance at this time resolution?  The reference monitors do not report data this frequently I don't think.

Authors: 4 of the FEM monitors report 1-minute data that we use for comparison. This has been noted in the text and is hopefully more clear with this revision

Line 511 - Which models, specifically?

Authors: The machine learning models have been clearly labelled in all Tables. Machine learning models correspond to Models 17 - 21

Line 529 - "appears to be" is qualitative and not useful; quantify the difference.

Authors: Thank you. We have done so. We have changed the sentence to read:

"Relative humidity at the co-located sites (three of the four co-location sites have a median RH close to 50 % or higher) is larger than at the other sites in the network (7 of the 12 other sites have a median RH < 50%) (**Figure 2b**)."

Line 549 - Why does Figure 2b appear to have a different shape to the box and whisker plots relative to the other parts of this figure?

Authors: This is because for Figure 3b, we are evaluating the performance metrics across different 3-week periods of data left out. In Figure 3b we are evaluating transferability across space not site.

We have noted the following in the text:

In section 2.3.4:

"To evaluate how transferable the calibration technique developed at the co-located sites was to the rest of the network, even after conducting LOSO CV, we left out each of the five co-located sites in turn and using data from the remaining sites ran the models proposed in **Tables 2** and **3**. We then applied the models generated to the left-out site. We report the distribution of RMSE from each calibration model considered at the left-out sites using box-plots (**Figure 3**). For correction C1, we also left out a three-week period of data at a time and generated the calibration models based on the data from the remaining time periods at each site. For the machine learning models (Models 17 – 21), we used CV = LOBD. We plotted the distribution of RMSE from each model considered for the left-out three week period (**Figure 3**)."

Lines 553-554 - confusing sentence

Authors: Thank you. We have made the following change to the sentence:

"Large reductions in RMSE are observed when applying simple linear corrections (*Models 1 - 4*) to the uncorrected data across C1, C2, C3 and C4. Increasing the complexity of the model does not result in marked changes in correction performance on different test sets for C1 and C2."

Figure 2 caption - typos: no (d) or (e)

Authors: Thank you for catching this mistake. We have updated the caption

Figure 3 - Need better labels in the y-axis for the models; they are referred to as "Model 1, 2, …" in the caption but differently on the figures themselves.

Authors: Thank you. We have updated the caption to make this more clear

Line 625 - "It appears" is qualitative language and not helpful. Don't you quantify the variation later in the sentence? You should prove that these variations are significant and then leave no doubt to the reader what the conclusion should be.

Authors: Thank you. We have updated the sentence to the following:

"There is larger temporal variation (max 32.79 µg/m$^3$), in comparison to spatial variations displayed across corrections (max: 11.95 µg/m$^3$). Model 16 generated using the C3 correction has the greatest spatial and temporal RMSD in comparison with all other models. Models generated using the C3 and C4 corrections displayed the greatest spatial and temporal RMSD vis-a-vis C1 and C2."

Lines 626-627 - These max numbers look like they are all due to one specific model (there is one row and one column with dark green colors, while all other models are pink). If you take out this one model, does your conclusion hold? Why is the one model so different than the others?

Authors: The reviewer is correct that one specific model using Model 16, C3 correction, that we have earlier showed (Table 2) doesn't work too well, resulted in the greatest RMSD. We do not remove this model from our ensemble because it is a widely used correction model. The metrics we display: uncertainty and normalized range capture the variation introduced by the choice of the calibration equation. We have updated the paper to make this more clear.

Line 652 - Was "exposure assessment" used/defined earlier? I don't know if this is something new calculated from the PM concentrations or not.

Authors: Thank you. We have used 'corrected measurements' instead of the term 'exposure assessments

Lines 705-707 - confusing sentence

Authors: Thank you. In order to ensure our language was consistent we have updated the sentence to read as follows:

"We also found that the calibration models yielded different performance results in different pollution regimens. Machine learning models developed using C1, and models developed using C2 were better than multivariate regression models generated using C1 at capturing peaks in pollution (> 30 µg/m$^3$ ). All models using C3 and C4 yielded poor

performance results across both concentration ranges ($PM_{2.5} > 30$ μg/m$^3$ and $PM_{2.5} \leq 30$ μg/m$^3$) (**Tables 4** and **5**)"

Lines 745-746 - redundant wording

Authors: Thank you. We have updated the sentence to read as follows:

"The normalized range in the corrected measurements, on the other hand, was large; however, most of the corrected measurements fall within a relatively small interval. Thus, deciding which calibration algorithm to pick has important consequences for decision-makers using data from this network."

Technical corrections

Line 234 - missing space

Authors: Thank you. We have changed this.

Lines 283 and 333 - single sentence paragraph

Authors: Thank you. We have changed this.

Line 495 - 'correction'

Authors: Thank you. We have changed this.

Section 3.1 appears twice, all succeeding sections need to be renumbered.

Authors: Thank you. We have changed this.

Line 542 - this sentence is not needed

Authors: Thank you. We have deleted this sentence.

Line 546 - "LOBD"

Authors: Thank you. We have changed this.

Line 608 - inconsistent ways of referring to months

Authors: Thank you. We have changed this.

Line 692 - why the colon?

Authors: Thank you. We have removed the colon

Misplaced or missing commas - Lines 684, 685, 702, 711, 751

Authors: Thank you. We have changed this.

---

## Author Response (AR2)

Overall, I commend the authors for significantly rewriting portions of the manuscript to make the whole thing more clear and accessible. I still think that there is much improvement that could be done further, as my comments below will point out. Some of my comments are regarding style moreso than technical content, so final decision on publication should be between the authors and the journal.

Authors: Thank you

One concern I have is that the reviewer comments from the first round were not fully implemented nor highlighted in the track changes version. Thus, it was hard to see if some comments were address satisfactorily or not.

Some examples:
Reviewer 1 - figure resolution: It does not look to me that any of the figure resolutions were improved.

Authors: The figures included in the paper are of high resolution

Reviewer 1 - Abstract and L49: The abstract was addressed but L49 did not change at all (it is now L58).

Authors: We removed stating that reference monitors were 'gold standard' in the Abstract. We left this sentence in the Introduction but have now removed it in this new iteration.

Reviewer 1 - L119: While technically this comment was addressed, you also switched from using the "coefficient of determination, R^2," to "Pearson correlation coefficient, R," without any explanation or indication in the tracked changes version of the document.

Authors: We have now used Pearson correlation coefficient throughout

Reviewer 1 - L240: There is no new discussion in the paper, as was asked for. The response is just cut and pasted from the original manuscript.

Authors: We had included the following sentence to address the reviewer's concern about using non-independent meteorological parameters:

"As T, RH and D are not independent (**Figure S8**), the 16 linear regression models include adding the meteorological conditions considered as interaction terms, instead of additive terms. The remaining 5 relied on machine learning techniques."

Reviewer 1 - L333: I think the proper symbol is still not being used in the equation.

Authors: We have made this change everywhere

Reviewer 1 - L457: The text you pasted in the response does not match what is in the manuscript, now beginning at L574.

Authors: We had made a typo in our response which differed from the text in the main manuscript.

Reviewer 1 - L472: The text you pasted in the response does not match what is in the manuscript and is not highlighted; I think this is now L796.

Authors: We had made a typo in our response which differed from the text in the main manuscript.

Reviewer 1 - L528: This detail, about the sensor being in the shade, was not added to the manuscript but is important enough that it should be, now near L524.

Authors: Thank you. We edited the sentence to read as follows:

"The sensor CS19 is the only one that recorded lower temperatures than those at any of the other sites likely due to it being in the shade."

Reviewer 2 - "For better readability of the final paper..": Your response looks cut off; your sentence ends in "and". As I say above, I appreciate your work put into restructuring the paper; the manuscript is certainly better. Maybe it is just the way the tables are formatted in the draft stage, but it doesn't look like anything was done to improve the table readability.

Authors: We reorganized the tables in our last revision

Reviewer 2 - comment about "it" and "this": There are still about 15 "this"'s and about 6 "it"'s that are used as subjects of a sentence.

Authors: Thank you. We have tried to eliminate as many of these as possible.

Reviewer 2 - Line 335: It's not clear what is new as a result of addressing this comment.

Authors: We had reorganized the manuscript considerably including the old Figure S9

Reviewer 2 - Line 705: Should be "regimes", not "regimens". In the paper this is L873.

Authors: We have made this change

New comments based on the latest manuscript (with new line numbers):

L251 - What type of correlation is used here? I'm confused because of the switch from the first submission to this one with no explanation.

Authors: We have clarified that this is the Pearson correlation coefficient in this revision

L270 - What instrument errors add uncertainty for 1-minute data but makes hourly data more certain?

Authors: We have clarified that the error is due to the finer time resolution:

"..likely due to the impact of passing sources. It is also important to mention that minute-level reference data may have some additional uncertainties introduced due to the finer time resolution"

Section 2.3 - I think this section is new (though its not highlighted as such), and I appreciate the attempt to better outline the paper. I wonder now, though, if there is too much repetition in the paper now. This section of three lists are essentially the section headers 2.3.1 - 2.3.7 and 3.1.1 - 3.1.7 (sections 2.3.8 and 3.1.8 also repeat). Is this much redundancy necessary? Also, most are exact repeats but some are not exact, which makes me wonder why. If you are going to repeat, be consistent with the wording and make these lists in Section 2.3 have a numbering system that matches with the rest of the paper.

Authors: Thank you. We have compressed all of section 2.3 and subsections to read as:

Section 2.3.2 - An entire section of just one sentence is weird to me. With the previous comment, I don't think this is necessary.

Authors: Thank you. We have got rid of this subsection.

L433 - should reference Fig 4

Authors: Thank you. We have added this reference

L536 - be specific, which sensors? Why?

Authors: Thank you. We have tried to be more specific:

"RH values at co-located sites during C3 and C4 tend to be higher than conditions experienced by Love My Air sensors: CS8, CS10, CS15, CS16, CS17, CS18, CS20 likely due to the different microenvironments experienced by the different sensors."

Fig 2 - "a)" and "b)" are in odd locations, hard to read
Authors: We have adjusted the position of these labels

L596 - This paragraph looks mostly like information that should be in the figure caption and not repeated in the text.

Authors: Thank you. We have removed this paragraph

L612 - "evidenced" instead of "evinced"

Authors: Thank you. We have made this correction

Section 3.1.5 - Are both R and RMSE needed to get your points across?

Authors: Thank you. We have removed reporting R

L683 - Should this actually be higher? I'm not sure I understand how this relates to L668-669. Figs 7 and 8 aren't high enough resolution to figure this out. Either way, why? I don't think Section 4 addresses why.

Authors: Thank you. We have removed reporting R

L711 - Why? I don't think Section 4 addresses why.

Authors: Thank you. We have addressed this with the following sentence:
"We note that although the uncertainties in the data are small, the average normalized range tends to be quite large likely due to outlier corrected values produced from some of the more complex models evaluated using the C3 and C4 corrections."

L731 - should reference Fig 10

Authors: Thank you we have corrected our figure numbers.

L733 - this should be Figure 10, not 9

Authors: Thank you we have corrected our figure numbers.

Fig 10 needs bigger fonts to be readable. There is also a stray sentence below the x axis label. As with all other figures, the resolution is low.

Authors: Thank you we have removed the stray sentence.

Section 4 - By this point in the paper, I have forgotten what C1, C2, C3, and C4 are. A quick reminder/summary might be helpful (many readers may skip to this section of the paper anyways).

Authors: Thank you. We have added this reminder in the paper.

L780 - missing comma after seasonality

Authors: Thank you we have added this

L794 - what other potential factors?

Authors: Thank you we removed this phrase

L856 - "demonstrated,"

Authors: Thank you. We have made this correction.

To summarize my thoughts on organization, it looks like Section 2 is on methods of data manipulation, Section 3 is stating what is plotted (with no actual analysis), and Section 4 is where the analysis is supposed to take place. This organization leads to a lot of repetition, but some inconsistent repetition. For example, Section 4 does not actually address each subsection found in Sections 2 and 3, nor does it reference each figure and table like the previous sections do. (Because all figures and tables are referenced in Section 2 and 3, it might be a bit difficult to figure out figure placement when the paper is typeset.)

Authors: Thank you. Section 3 comprises of Results and Section 4 comprises of the Discussion.

I would think, at the very least, that Sections 3 and 4 can be combined in some way and that an analysis is presented for each figure and table. Some of what is currently in Section 3 can go in the Figure captions or in Section 2. A new Section 4 can be a short summary of the take-home conclusions from the study.

Authors: Thank you. We have tried to implement as many of these suggestions as possible.

These are my suggestions to make the paper read better. Overall, I think the science and technical content that are here are useful, but it is hard to pick out the conclusions  in the current format. Some of that might just be formatting the current paper and typesetting in the journal format will help, but it is hard to say that for certain.